# Do bacterial viruses affect the framboid-like mineral formation?

Paweł Działak[1], Marcin D. Syczewski[2], Kamil Kornaus[3], Mirosław Słowakiewicz[2], Łukasz Zych[3], Andrzej Borkowski[1]

[1]Faculty of Geology, Geophysics and Environmental Protection, AGH University of Science and Technology, Al. Mickiewicza 30, 30-059 Krakow, Poland.
[2]Faculty of Geology, University of Warsaw, ul. Żwirki i Wigury 93, 02-089 Warsaw, Poland.
[3]Faculty of Materials Science and Ceramics, AGH University of Science and Technology,
Al. Mickiewicza 30, 30-059 Krakow, Poland.

*Correspondence to*: Paweł Działak (dzialak@agh.edu.pl)

**Abstract.** Framboidal pyrite has been a matter of interest of many studies due to its abundance in a wide range of environments, and as a marker of redox conditions. However, the clear origin of framboidal pyrite remains unresolved. The studies are preliminary laboratory investigations on the influence of the shape and physicochemical properties of bacteriophages on the synthesis of framboid-like structures. The paper discusses the possible role of bacteriophages (bacterial viruses) in the precipitation of sulphide minerals (FeS and CuS) and their impact on the formation of framboid-like structures. Here, two bacteriophages (*Escherichia virus P1* and *Pseudomonas phage Φ6*), which differ significantly in shape and physicochemical properties, were used. Our observations suggest that viruses may bind ions from the solution. Moreover, we showed that bacteriophages P1 can lead to the formation of finer mineral particles of FeS and CuS, whereas the framboid-like structures were found only in experiments with precipitation of FeS. However, the lipid-enveloped *Pseudomonas phage Φ6* did not cause the formation of similar structures. It is assumed that *Escherichia phage P1* can promote the formation of the FeS-based framboid-like or spherical structures. The proposed four-step conceptualized mechanism facilitating the framboid-like structure synthesis via viruses is as follows: (i) binding of ions by capsids, (ii) bacteriophages behave like crystallisation surface, (iii) destabilisation of the colloid (ζ-potential ~ 0), (iv) formation of fine agglomerates and subsequent formation of small crystallites. Further studies are required to find all factors that may be affected by bacteriophages during sulphide precipitation. In addition, it is important to consider viruses present in sedimentation environments, despite possible difficulties in laboratory culturing. The consideration of such viruses may make laboratory testing more valid in terms of sedimentation environments.

## 1 Introduction

### 1.1 Bacteriophages and their presence in various environments

Bacteriophages are present virtually in all environments on Earth (Sharma et al., 2017). They are present even under unfavourable conditions such as deep-sea hydrothermal sediments (Liu et al., 2006), deserts (Prestel et al., 2008), glaciers (Bellas and Anesio, 2013) or hydrothermal veins (Lossouarn et al., 2015; Millard et al., 2016). Many biogeochemical cycles

can be affected by viruses due to their possibility of infecting bacteria (Suttle, 2007). Bacteriophages can cause mainly two pathways of infection: lytic and lysogenic. The lysogenic cycle allows for the replication of viruses within the genome of the bacterium but without the lysis of the host. On the other hand, the lytic cycle produces infectious bacteriophages and causes host lysis (Hobbs and Abedon, 2016). The lytic cycle seems to be more important from the point of view of microbial communities, because bacteriophages can control the number of host populations and thus influence ecosystems (Weitz and Wilhelm, 2012). The abundance of viral particles in water or sediments can exceed the number of bacteria by orders of magnitude (Fuhrman, 1999; Steward et al., 1996). The number of bacteriophages is estimated to be $10^4$ to $10^8$ per mL in aquatic environments (Wittebole et al., 2014). The total number of bacteriophages is estimated at $10^{31}$ (Weitz and Wilhelm, 2012).

Many viruses have a symmetric, crystal-like structure that can be analysed with X-ray diffractometry. Most of them have an icosahedral capsid (virus head) without an outer lipid membrane. Viruses studied here have different shapes and diameters. *Pseudomonas phage Φ6* has an icosahedral shape (60 nm in diameter) and a lipid envelope (Vidaver et al., 1973), while *Escherichia phage P1* has an icosahedral shape (75 nm in diameter) with a long tail (220 nm in length) (Fermin et al., 2018). Bacteriophages have a surface charge and potential interactions with ions present in solutions can be assessed based on ζ-potential measurements. Therefore, if the relevant conditions for mineralization occur (e.g., exceeding the solubility equilibrium), the process should be visible in changes of the ζ-potential. Considering the importance of viruses in biogeochemical processes, bacterial viruses have not been studied so far as crystallisation factors in sulphide environments, especially their involvement in the formation of framboid-like structures.

**1.2 Microorganisms and their role in the formation of minerals.**

Currently, microorganisms are known to play a pivotal role during the formation and precipitation of minerals. Bacterial and archaeal activity can lead to a change in physicochemical conditions in various environments, under both aerobic and anoxic conditions. For example, biochemical activity can lead to the formation of various biominerals (Lowenstam, 1981) and mechanisms responsible for their formation are termed biomineralization (Lowenstam and Weiner, 1989; Mann, 1988). Microbial activity can also strongly affect the oxidation of sulphide and Fe(II) from pyrite and other sulphide minerals producing sulphate, Fe(III) minerals and protons, leading to a decrease in pH leading to the formation of acid mine drainage (Chen et al., 2016). On the other hand, sulphate-reducing bacteria can produce sulphide under anaerobic conditions (Muyzer and Stams, 2008) and this process was crucial in some circumstances in the formation of various metal ores.

**1.3 Framboids and their origin**

In certain processes, sulphide and oxide minerals can be shaped as framboid structures. The framboid is a commonly known micromorphological feature of some sedimentary minerals, e.g., framboidal pyrite (Wilkin and Barnes, 1997) and framboidal magnetite (Hua and Buseck, 1998). The creation of framboidal pyrite or framboidal greigite (a precursor of pyrite) can be linked (i) with reducing conditions in sediments where $H_2S$ is produced, and (ii) with euxinic water where sulphides and no-oxygen conditions coexist (Popa et al., 2004). Hence, the pyrite framboids are markers of the redox transition between oxygen-containing and anaerobic/sulphidic waters. However, it is not clear how the framboidal structure is created. Some experiments conducted under laboratory conditions showed that framboid-like pyrite can be synthesized abiotically (Ohfuji and Rickard,

2005; Farrand, 1970; Graham and Ohmoto, 1994; Sweeney and Kaplan, 1973; Wilkin and Barnes, 1996). Moreover, the hydrothermal origin of framboidal pyrite has also been postulated (Wilkin and Barnes, 1997). Nonetheless, these structures may be formed through microbial activity (Popa et al., 2004). However, an additional possible factor, which should be discussed is involvement of bacteriophages (bacterial viruses). Is it possible that viruses can induce or influence the formation

of framboid-like structures during mineral precipitation? This question seems to be open due to the lack of sufficient experimental proofs. However, it should be stressed that bacteriophages have been reported as factors affecting mineral formation processes. Several studies showed that viruses can be used as a modifying factor that affects the precipitation of various nanomaterials (Ivanovska et al., 2004; Nam et al., 2004; Slocik et al., 2005). In laboratory experiments, bacteriophages were also shown to likely influence vaterite formation – a metastable form of calcium carbonate (Słowakiewicz et al., 2021).

A few studies have been conducted so far in relation to modern siliceous (Daughney et al., 2004; Laidler and Stedman, 2010; Peng et al., 2013), iron-rich (Kyle et al., 2008; Orange et al., 2011),  carbonate (Pacton et al., 2015; Słowakiewicz et al., 2021; Perri et al., 2022), and microbial mat (De Wit et al., 2015; Perri et al., 2018; Pennafirme et al., 2019; Pacton et al., 2014; Carreira et al., 2015) environments. Therefore, in light of the above-mentioned works, bacteriophages might play a role in various sedimentation processes. Given the fact that (i) bacteriophages have crystal-like structures of capsids; (ii) the

dimension of capsids is very small (about 100 nm); (iii) capsids may have a different structure, the question about virus-origin of framboids seems to be well-founded. However, the involvement of bacteriophages may be only one of the various processes that lead to the formation of framboids.

Therefore, in this work, we demonstrate experiments that test the hypothesis that framboid-like structures can be created during mineral precipitation in the presence of bacterial viruses under laboratory conditions. For the analysis of minerals obtained,

epifluorescence microscopy, scanning electron microscopy (SEM), and X-ray diffraction (XRD) were used. Moreover, $\zeta$-potential measurements were conducted in order to check: (i), the electrochemical properties of bacteriophages; (ii) the changes in the electrochemical properties of minerals in the presence of viruses. For obvious reasons, we focussed on sulphide minerals, but we did not assume that framboid-like structures cannot be formed by precipitation of other minerals. The formation of sulphide framboids in the presence of viruses can be extremely important for the correct interpretation of the presence of such

structures in sediments and aquatic systems with the developed euxinic zone.

## 2 Materials and Methods

### 2.1 Preparation of bacteriophages and experimental setup

The overall scheme of the experiments is presented in Figure 1.

### 2.1.1 Preparation of bacteriophages

*Escherichia phage P1* and *Pseudomonas phage* Φ6 were obtained from the Leibniz Institute DSMZ-German Collection of Microorganisms and Cell Cultures. Three media were prepared: bottom agar – tryptic soy broth (TSB) containing 1.5 % agar; top agar - TSB containing 0.75 % agar; and liquid TSB without agar. The media were sterilised at 121°C for 30 min and cooled down to 50°C. Bottom agar was poured onto the Petri dishes, and plates were dried to remove the condensed vapour on the

surface of the solid medium. It is crucial to dry plates before pouring the top agar, e.g., in a laminar flow cabinet. Otherwise, the top agar layer will not adhere to the bottom agar. Sterile 1M MgSO$_4$ solution was added to the cooled liquid TSB and the final concentration was set at 5 mM. To a 15-mL tube with 0.1 mL of diluted bacteriophage solution, 1 mL of liquid TSB containing 5 mM MgSO$_4$ and 0.1mL of an overnight (18 h) liquid bacterial culture was added (OD$_{550}$ ranges for *Escherichia coli* and *Pseudomonas syringe* were 0.7-0.9 and 0.4-0.5, respectively). Cooled top agar (4 mL) containing 5 mM MgSO$_4$ was added to the bacterial solution. The tube was gently mixed, and the solution was immediately poured onto a bottom agar plate. Solidified plates were incubated for 24 h at 37°C and 48 h at 25°C for the *E. coli* and *P. syringe,* respectively. The latter cannot be overheated because high temperature prevents their growth.

### 2.1.2 Purification of bacteriophages

The top agar was scraped using a glass rod and transferred to a 50 mL polypropylene tube. A 20 mL of sterile bacteriophage buffer (MgCl$_2$ 20 mM and Tris-HCl 20 mM) was used and the mixture was thoroughly mixed using vortex. Optionally, 3 mL of chloroform can be added to the *Escherichia phage P1* culture to facilitate the removal of bacterial debris. Importantly, chloroform destroys the lipid envelope of *Pseudomonas phage* Φ6, therefore the solvent cannot be used. The mixture was centrifuged at 4400×g for 5 min to remove agar.

The supernatant was transferred to 2 mL polypropylene tubes. The tubes were firstly spined at 13000×g for 5 min to remove the bacterial debris. The supernatant was transferred to 2 mL tubes. The tubes were centrifuged at 24000×g for 45 min at 5°C. The supernatant was discarded and the tubes with pellet were centrifuged to remove the remaining liquid. Viral pellets were visible with a naked eye.

1 mL of a sterile 0.9 % NaCl solution was added to the first tube and the pellet was resuspended. The whole solution was transferred to the next tube and used to resuspend the pellet. This step was repeated until all pellets in the tubes were resuspended. The solution was placed in a 15-mL polypropylene tube and 14 mL of sterile 0.9 % NaCl was added. The solution can be stored in the fridge at 5°C for up to one month.

### 2.1.3 Quality control

Viral solution (0.1 mL) was mixed with 10 µL of the previously diluted (10000×) Sybr®Gold dye (Thermo Fisher Scientific, USA). The solution was placed on a microscopic slide and examined using an epifluorescence microscope with the blue filter (DM500 filter with a band-pass 460-490 nm excitation filter). The count of bacteriophage particles was estimated using plate techniques with serial dilution of bacteriophages suspension obtained after purification step. Additionally, the UV-related empirical formula was used to quickly assess the density of bacteriophage suspension:

$$\frac{virions}{mL} = \frac{(A_{269} - A_{320}) \cdot 6 \cdot 10^{16}}{number\ of\ bases/virion} \text{ (Day and Wiseman, 1978)}$$

In all experiments, the density of bacteriophages suspension was normalised to $10^{10}$ mL$^{-1}$.

### 2.1.4 Precipitation experiments

Solutions of FeSO$_4$ (Avantor, Poland), CuSO$_4$ (Chempur, Poland), and Na$_2$S (Avantor, Poland) (0.2 M) were prepared. A beaker with 10 mL of FeSO$_4$ or CuSO$_4$ solution was thoroughly mixed on a magnetic stirrer (stirring condition) or without stirring. Importantly, iron sulphide precipitation was carried out under an oxygen-free atmosphere (chamber Secador Techni-Dome 360). The oxygen concentration in the chamber was controlled by the Greisinger GOX 100 oxygen sensor (measure range 0 – 100%; accuracy 0.1%). The chamber was flushed with pure nitrogen (Multax, Poland, 99.999%) until the concentration of oxygen was below 0.5%. The water prior to use was degassed by autoclaving (121°C, 15 min). The bacteriophage solution (1 mL) and the subsequent Na$_2$S solution were poured according to two variants: (i) 2 to 20 µL of Na$_2$S for ζ-potential and size distribution measurements; (ii) 5 mL of Na$_2$S (in 0.5 mL portions) for SEM and XRD analysis. For SEM and XRD analyses, the obtained precipitates were centrifuged, the supernatant was discarded, and the pellet was washed in pure water. Eventually, the pellet was resuspended in 10 mL of analytical grade acetone for subsequent XRD measurements (Fig. 1). Washing in acetone allows rapid removal of water residues, drying, and transfer to capillaries for XRD.

## 2.2 ζ-potential measurements and Z-average size distribution

ζ-potential and Z-average size distribution measurements were obtained using a Zetasizer Nano-ZS (Malvern).

### 2.2.1 Isoelectric point of bacteriophages

To find the isoelectric point of bacteriophages, a series of measurements was carried out in a phosphate buffer with pH ranging from 4.5 to 8. The bacteriophage solution (1 mL) was mixed with 14 mL of buffer at the fixed pH (4.5; 5.6; 6.2; 7.0; 7.5) and then placed in a measurement cell.

### 2.2.2 Changes of the ζ-potential of bacteriophages in presence of substrate solutions

Measurements were carried out to assess the changes of ζ-potential of bacteriophages as an indicator of possible interactions with metal cations and anions used in precipitation experiments. The solution of bacteriophages (1 mL) was mixed with 5 mL of FeSO$_4$ (0.2 M), CuSO$_4$ (0.2 M), or Na$_2$S (0.2 M). Control samples were pure bacteriophages (1 mL in 0.9% NaCl) diluted with 5 mL of deionised water. The samples were incubated for 3 to 5 min. Subsequently, the solutions were transferred to measurement cells and measured.

### 2.2.3 Measurements of sulphide precipitates

ζ-potential measurements were performed to check possible changes in ζ-potential of precipitated minerals in the presence of viruses. The reaction mixture was placed in a measurement cell after the precipitation as described in Section 2.1.4 in variant (i). Additionally, the Z-average size distribution measurements were conducted. However, due to strong agglomeration, the obtained results may exceed the measurement range of the device.

## 2.3 XRD

The samples were taken out of the chamber and, after the acetone evaporation, immediately transferred to glass capillaries and sealed. The X-ray diffraction analysis was performed with a Malvern Panalytical X'Pert PRO MPD (Malvern Panalytical, Malvern, UK) diffractometer. The registration was carried out with a CoKα radiation source at 40 kV and 30 mA in the range of 4–84$^O$ 2θ with a step of 0.0260$^O$ 2θ. The results were analysed using X'Pert Plus HighScore software with access to the Crystallography Open Database.

## 2.4 Electron microscopy

Microscopic analyses were conducted using a Carl Zeiss AURIGA (Carl Zeiss Microscopy GmbH) field emission scanning electron microscope (SEM) coupled with two energy-dispersive spectrometers XFlash 6|30 (Bruker Nano GmbH). Crystallite samples were placed on the glass slide and coated with 20 nm layer of carbon by a vacuum coater (Quorum 150T ES). Analysis was done with a 20 kV acceleration voltage, a 120 μm aperture, and the working distance at approximately 10 mm. The bacteriophage characterisation was conducted using a scanning transmission electron detector on the same microscope. A drop of purified bacteriophages was applied on a TEM mesh covered with a carbon foil with a thickness of 3 nm. Next, they were contrasted with 1% solution of uranyl nitrate for 2 min and subsequently dried out. Finally, they were coated with 8 nm of chromium layer by vacuum coater. Analysis was done with bright field mode at 30 kV acceleration voltage, a 120-μm aperture, and the working distance at approximately 10 mm.

## 3 Results

The procedure of obtaining pure bacteriophage suspensions was highly efficient. The obtained bacteriophages were clearly visible after Sybr®Gold staining due to the agglomeration occurring especially when the density of bacteriophages suspension exceeded $10^{10}$ mL$^{-1}$. Furthermore, the obtained suspension was free from bacterial debris, as can be seen in the epifluorescence microscopic and SEM images (Fig. 2). The yield of bacteriophages was in the range of $1\times10^{8}$ to $1\times10^{11}$ bacteriophages mL$^{-1}$ and before the next experiments bacteriophage suspensions were normalised to $10^{10}$ mL$^{-1}$.

### 3.1 ζ-potential measurements of bacteriophages

The measurements have revealed that ζ-potential is different for studied bacteriophages and is pH-dependent (Fig. 3a). Under the tested pH, it has not been possible to obtain the ζ-potential value of zero and thus determine the isoelectric point. For the pH range of 5.6 to 7.5, the ζ-potential for both bacteriophages were considerably stable, ranging from -5 to -15 mV. At pH 4.5, the ζ-potential of P1 reached -27 mV, while for Φ6 the potential reached -2 mV.

### 3.2 ζ-potential measurements and Z-average size distribution of precipitates

ζ-potential measurements of FeS precipitates (Fig. 3b) have shown that there are differences in the samples (groups of the same cation) with and without bacteriophages. Data analysis has shown that the differences between samples were statistically significant (**). It should be noted that ζ-potential in the attempt of CuS (Fig. 3b) with the bacteriophage P1 has changed to positive values (+1mV). The conductivity was not significantly different in all samples and ranged from 15 to 16 mS/cm (Fig. 3b). The conductivity was measured for control purposes to check whether the ionic strength was similar in all samples.

ζ-potential of both bacteriophages at pH 7 in phosphate buffer was below -10 mV (Fig. 3a). The ζ-potential measurements of the bacteriophages with the addition of FeSO$_4$, CuSO$_4$ or Na$_2$S were conducted in order to check whether the ions can change the surface charge of the bacteriophages possibly indicating the binding of the ions. For both bacteriophage suspensions in a 0.9 % NaCl solution prepared as control, ζ-potential was approximately -5 mV (Fig. 3c). For bacteriophage P1, the addition of sole FeSO$_4$ or CuSO$_4$ caused a slight, but significant change (-3.3 and -3.7 mV respectively). However, addition of sole Na$_2$S increased the potential significantly (-15.8 mV). In case of bacteriophage Φ6, ζ-potential was -3.1 mV. The addition of

FeSO$_4$, CuSO$_4$ caused only an insignificant change (-0.9 and -1.22 mV, respectively). However, addition of sole Na$_2$S increased the potential significantly (+11.9 mV).

The measurement of Z-average size distribution (Fig. 3d) has revealed that addition of bacteriophages to the reaction mixture significantly changed the size of precipitated FeS and CuS particles. There has been an increase of 275% (2000 nm - 5500 nm) and 405% (2000 nm - 8100 nm) for FeS+P1 and FeS+Φ6 respectively. For CuS+P1 and CuS+Φ6, addition of bacteriophages

caused the formation of smaller particles. There has been about 70% decrease in size of precipitates (5900 to 4500 nm and 5900 to 4000 nm respectively).

### 3.3 XRD

The X-ray diffraction results are presented in Figure 4. Phase composition of samples did not differ significantly. The strongest signals derived from minerals resulted from oxidation of samples during the grinding and preparation of samples for XRD

analysis. In copper sulphide synthesis, traces of covellite were found. However, its identification was hampered by oxidation products, other synthesis products, and residual substrates used in the synthesis. The XRD results confirmed the presence of kröhnkite, chalcanthite, natrochalcite, butlerite, and jarosite (probably oxidation products). Traces of pyrite and probably troilite were noted in the synthesis of iron sulphides. Like in copper sulphides, the analysis of iron sulphides did not differ significantly probably due to the oxidation of samples. The results additionally proved the presence of mohrite (rest of the

substrate).

### 3.4 Morphology of FeS and CuS precipitates

In order to check whether the observed minerals were ferrous and copper sulphides, the EDS spectra were obtained for every observed sample. The EDS spectrum should not contain significant signals indicating oxygen and sodium in the sample. Such elements could indicate the impurities from reagents or products after reaction, and the observed crystals probably were

sulphates. In Figure 5 samples of EDS spectra of minerals observed under SEM are presented. Spectra 1 and 2 revealed clearly visible signals from iron and sulphur. Neither oxygen nor sodium in significant quantity were noted. Such spectra suggest that the observed minerals were ferrous sulphides. If these spectra revealed strong signals of sodium and/or oxygen in relation to sulphur signal (e.g. spectrum 3), the observed minerals were not further analysed. In case of copper sulphide, the same approach was used. Spectra 4 and 5 contain strong signals of copper and sulphur. Oxygen and sodium were hardly detected. Hence, it

can be assumed that the observed minerals were copper sulphides. Spectrum 6 revealed strong signals indicating sodium and probably minerals contained sulphates.

### 3.4.1 Experiment with P1 bacteriophage and FeS precipitation

The obtained precipitates of FeS with and without P1 bacteriophage under both stirring and without stirring conditions were examined under SEM to analyse the morphology and dimension of the single and agglomerated/aggregated particles (Fig. 6).

The stirring conditions during our experiment caused the precipitation of quite massive structures (Fig. 6 a-d). In experiments with bacteriophages the similar structures were noted (Fig. 6, e-h), however much smaller particles (Fig. 6 f,g) were additionally noted in comparison to control experiments (Fig. 6d).

Furthermore, very small particles with dimensions close to 200 – 300 nm were found in great numbers (Fig. 6 h), while similar particles were not observed in control experiments (the smallest particles were approximately 0.8 – 1 μm; Fig. 6d). It should

be emphasized, that the framboid-like structures were not found in both experiments conducted under stirring conditions.

It would seem that a similar situation was observed in experiments under non-stirring conditions (Fig. 6, control: i–l; with bacteriophages: m–p). In experiments with bacteriophages, much smaller particles building larger agglomerates/aggregates were observed (Fig. 6, m-o) in comparison to the control sample (Fig. 6, i-k). Very small particles were also observed (Fig. 6p). However, the precipitated minerals with bacteriophages under non-stirring conditions revealed spheroidal structures that

were composed of finer particles (Fig. 6 m-o). These spheroidal structures resemble framboid-like structures and were not observed in control experiments. The diameter of these structures ranged from about 1 μm to more than 10 μm.

### 3.4.2 Experiment with Φ6 bacteriophage and FeS precipitation

Analogous experiments were conducted with Φ6 bacteriophages (Fig. 7) to check whether the bacteriophage capsids enveloped by lipid membrane can cause similar effects as in P1 bacteriophages. It should be noted that Φ6 phage caused the

overrepresentation of smaller mineral particles similar to P1 bacteriophage. However, neither stirring nor non-stirring conditions caused the precipitation of framboid-like structures. Only the potentially spheroidal structures could be seen (Fig. 7c), but these structures were not as clear as in experiments with P1 phage. Generally, the images obtained from experiments with Φ6 phage were not different from those obtained from control experiments (without bacteriophages).

### 3.4.3 Experiment with bacteriophages and CuS precipitation

It was also interesting to check whether the framboid-like structures may appear in presence of bacteriophages in case of non-ferrous sulphides. The copper sulphide can be precipitated much more easily in comparison to ferrous compounds, especially due to its higher stability under oxygen atmosphere. The experiments with copper compounds and P1 bacteriophages were conducted under stirring and non-stirring conditions, and the obtained minerals were examined under SEM (Fig. 8). The obtained results indicate that the presence of bacteriophages in the reaction mixture can strongly induce the creation of

agglomerates/aggregates that were built of smaller particles in relation to control experiments without bacteriophages. Similar observations were noted in non-stirring FeS experiments, but here it was especially visible in experiments under stirring conditions. The control precipitation created 'massive' agglomerates formed by large particles (Fig. 8 a-d). In contrast, in experiments with bacteriophages, the precipitates consisted of fine particles (Fig. 8 e-h). This difference was even clearly visible with a naked eye. CuS precipitates in suspension after precipitation in the presence of P1 bacteriophages had a fine,

sand-like texture, while CuS precipitates without bacteriophages formed a more compact mass. It should be noted that the framboid-like structures were not observed in any CuS experiments.

### 4 Discussion

Viruses are at the border of the biotic and abiotic worlds. They do not carry any biological processes outside of living cells.

However, as with other biological structures, they contain several proteins with different electrochemical properties. Thus, based on the DLVO theory (Derjaguin, Landau, Vervey, and Overbeek theory) (Derjaguin and Landau, 1993; Verwey, 1948),

it was assumed that viruses may bind ions from the solution and thus alter local physicochemical properties. If the viruses bind to the precipitated mineral, then the net surface charge of the virus-mineral complex might be changed. Daughney et al. demonstrated that viral capsids can bind iron ions and nucleate iron minerals (Daughney et al., 2004). Our $\zeta$-potential measurements of viruses agree with the data provided by Daughney et al. Further, the $\zeta$-potential measurements in our study reveal that viruses have a pH dependent negative charge. The charge is stable at neutral pH. However, compared to the work of Daughney et al., here we used different types of bacteriophages (P1 and $\Phi$6) and different solutions (phosphate buffer), whereas Daughney et al. used PWH3a-P1 and 0.5M NaNO$_3$, respectively. Our assumptions further prove the hypothesis presented by Daughney et al. that viruses bind ions. $\zeta$-potential of capsids with the CuSO$_4$ and FeCl$_2$ addition of solutions of CuSO$_4$ as well as FeCl$_2$ remained very similar to the control sample (pure viruses). However, the addition of Na$_2$S drastically changed the $\zeta$-potential. It is assumed that P1 capsids (naked) tend to bind S$^{2-}$ ions, while $\Phi$6 capsids (enveloped) can bind Na$^+$. The lipid envelope of $\Phi$6 capsid includes a significant amount of phosphatidylglycerol, which contains negatively charged phosphate groups (Laurinavičius et al., 2004). This can be one of the features that caused Na$^+$ binding. The P1 bacteriophage contains only proteins that might contain more positively charged groups, and thus bind to S$^{2-}$ ions. Analysis of variance (ANOVA) showed that the addition of every solution to P1 phage (CuSO$_4$, FeCl$_2$, Na$_2$S) caused a statistical change. For $\Phi$6 phage, a significant change was obtained only in the measurement with Na$_2$S.

Biological factors such as bacteria and possibly viruses can influence the precipitation of minerals, leading to the appearance of minerals with a different crystal structure. An example is the formation of greigite and mackinawite as a result of the activity of sulphate-reducing bacteria (Picard et al., 2018). In our experiments, it has been difficult to clearly see if the minerals produced may be of bacteriophage origin. The studies of iron sulphides precipitated under strictly anaerobic conditions may pose many problems in analytical processing. This is due to the rapid oxidation processes, which in effect masks any differences between biotic (with bacteriophages) and abiotic (control) samples. For example, it is not clear why troilite was observed. This is a high-temperature mineral, and it is not clear how bacteriophages or the experimental conditions would influence the precipitation of troilite. The presence of many phases resulting from the oxidation of precipitated sulphides causes difficulties in identifying individual minerals. For this reason, it was difficult to determine whether the bacteriophages had an influence on the formation of sulphides with a different crystallisation structure, compared to the sulphides formed in the control samples.

Although viruses have not been shown to alter the crystal structure of precipitated minerals, viruses appear to affect the surface properties of these minerals and the sizes of the crystallites. Measurements of FeS and CuS precipitates in the presence of bacteriophages revealed that the addition of viruses changed the $\zeta$-potential of mineral particles. There was an increase in the value of $\zeta$-potential in all samples. The charge was close to zero, and thus it caused destabilisation of the colloid. The statistical significance test showed that the samples FeS, FeS+$\Phi$6 and FeS+P1 are statistically different. Such differences were also found among CuS, CuS+$\Phi$6, and CuS+P1 samples. Statistical analysis confirmed our naked eye observations that the turbidity of the samples differed significantly. The precipitation experiment with bacteriophages has led to the formation of different structures, which formed visible agglomerates, whereas in control samples the phenomenon has not been observed.

The measured Z-average is the cumulative mean value of a set of particles measured by dynamic light scattering (DLS). The measurements showed that the size of FeS precipitates increased in samples with both types of bacteriophages, whereas the size of CuS precipitates behaved in the opposite way. This fact suggests that the average size of particles is bacteriophage-dependent, but also depends on the precipitated sulphide. However, the measurement might be erroneous, due to the dynamic processes (aggregation or agglomeration) that might occur in the measurement cell.

There is no exact definition of framboidal pyrite, making the evaluation of these structures difficult. These structures have been found, e.g., in sediments of euxinic basins (Wilkin et al., 1996); salt marsh sediments (White et al., 1990); coal (early peat formation) (Wiese and Fyfe, 1986); or carbonate environments (Kobluk and Risk, 1977). Similar structures have also been found within bacterial cells (Donald and Southam, 1999). The single-framboidal pyrite microcrystal can be shaped in different forms: cubic (Butler and Rickard, 2000), spherical (Wilkin et al., 1996; Berner, 1969), octahedral (Sweeney and Kaplan, 1973; Folk, 2005), pyritohedral (Murowchick and Barnes, 1987) or even icosahedral (Ohfuji and Akai, 2002). The diameter of framboidal pyrite can vary, depending on the environment (Table 1).

Table 1. Sizes of framboid diameter and their microcrystals. "-"– no data

| Microcrystal diameter [μm] | Framboid diameter [μm] | Source | Reference |
| --- | --- | --- | --- |
| 2-3 | - | Precambrian rocks with copper and lead-zinc ore; Mount Isa Shale | (Love and Zimmerman, 1961) |
| 0.12 | 12 | | |
| 0.9 | 10 | Deep-sea sediments; Angola Basin | (Schallreuter, 1984) |
| 0.7 | 12 | | |
| 2 | 24 | | |
| 0.3 – 0.7 | 3 - 10 | Super-anoxic fjord; South Norway | (Skei, 1988) |
| - | 1 - 2 | Coal basins; Bulgaria | (Kortenski and Kostova, 1996) |
| - | 50 - 70 | | |
| 1 | 10 - 15 | Mudstone; Lower Eocene, Marquez Shale | (Collins, 1982) |
| - | 30 - 80 | Muddy sediments (Miocene – Holocene) | (Ohfuji and Akai, 2002) |
| - | 5 - 20 | Modern reductive sediments | |
| 0.5 | 5 - 20 | Sulphur microbial mats; Kane Cave | (Folk, 2005) |
| 0.8 - 2 | 6 - 12.5 | Methane-derived carbonate chimneys; Gulf of Cadiz | (Merinero et al., 2009) |
| - | <200 | Sedimentary rocks of the gold deposits (Palaeozoic); Nevada, Victoria, USA | (Scott et al., 2009) |
| 0.3 - 5 | 3 - 10 | Sediments in the South Caspian Basin | (Kozina et al., 2018) |

The origin of framboidal pyrite remains the subject of debate (Kalatha and Economou-Eliopoulos, 2015). Based on the available literature, it seems that framboidal pyrite may have a different origin depending on the environment.

An overview of the environments in which framboidal pyrite occurs, shows that it can be formed in many different backgrounds: oxic, dysoxic, or euxinic (Wilkin et al., 1997). Furthermore, framboidal pyrite has also been synthesized

abiotically (Ohfuji and Rickard, 2005). Different approaches to the chemical synthesis of framboidal pyrite are presented in Table 2.

Table 2. An overview of the experimental conditions used to obtain framboid structures (literature data).

| Reagents | Temperature [°C] | Duration | Reference |
|---|---|---|---|
| $FeSO_4$, $H_2S$, $S^0$ | 65 | 2 weeks | (Berner, 1969) |
| $FeSO_4$, $H_2S$, $CaCO_3$; glycerine | 23 | Up to 1 year | (Farrand, 1970) |
| $FeCl_2$, $H_2S$, $S^0$ | 25, 60 or 85 | Up to 6 days | (Sweeney and Kaplan, 1973) |
| $FeCl_2$, $FeSO_4$, $Fe(NO_3)_3$, $Fe(NH_4)_2(SO_4)_2$, | 25; 100 | 2 days; 4 months | (Luther, 1991) |
| HCl, NaCl, FeS, $CaSO4$ | 150 - 300 | Up to 8 weeks | (Graham and Ohmoto, 1994) |
| Mackinawite or greigite, $H_2S$, | 70 | Up to 2 weeks | (Wilkin and Barnes, 1996) |
| $Na_2S$, $Na_2O_3Si$, $FeCl_2$, $Fe(NH_4)_2(SO_4)_2$, $Fe(NO_3)_3$ | 23 | Up to 2 years | (Wang and Morse, 1996) |
| FeS, $H_2S$, $KH_2PO_4/K_2HPO_4$; Ti(III) citrate | 60 - 100 | Up to 45 days | (Butler and Rickard, 2000) |

Each of the examples included in Table 2 required many reagents, or high temperatures, or a long time of the synthesis, which is entirely different from the conditions presented in this work. Here, we have also not included phenomena like aging.

Our synthesis of FeS-framboidal-like-structures was successful only in the trial with bacteriophages and without stirring. However, our framboid-like structures are very similar to the 'protoframboids' obtained by Wolthers (Wolthers, 2003; Wolthers et al., 2005) and Butler and Rickard (Butler and Rickard, 2000). The structures obtained by Wolthers are described

as protoframboids formed on the euhedral-pyrite overgrowth. Such structures are about 1 μm in diameter and are described as small and composed of poorly formed cubic pyrite microcrystals. Nonetheless, our experiments had different experimental conditions for the synthesis of pyrite, but it is likely that bacteriophages have led to the formation of similar structures (Fig. 9). However, based on our results, it cannot be concluded with certainty that the presence of bacteriophages led to the formation of framboids. Our research only showed that bacteriophages can influence the formation of spherical structures that may

resemble framboids and the appearance of finer crystallite fractions compared to controls. During the control experiments, the formation of agglomerated precipitates was observed, whereas in the presence of bacteriophages precipitation of very fine, non-agglomerated so massively crystallites was very clearly visible. In stirred conditions, we assume that the fluid currents were created and thus prevented framboid-like or spherical structures from forming. However, viruses can potentially be considered as factors that facilitate the formation of framboid-like structures. The bacteriophages selected for the study are not

necessarily common in pyrite sedimentation environments. However, bacteriophage P1 belongs to the order *Caudovirales*, which includes the most commonly occurring tailed bacteriophages in nature (Aishwarya et al., 2021). Furthermore, the icosahedral shape of bacteriophage P1 is one of the most common capsid forms of bacteriophages (Louten, 2016). On the other hand, bacteriophage Φ6 belongs to the group of viruses that infect common *Pseudomonas* bacteria. It is one of the

bacteriophages having a lipid envelope (Vidaver et al., 1973) and was chosen for comparative purposes. However, it should

be noted here that studies on the possibility of framboid pyrite formation in the presence of viruses should be continued with other bacteriophages present in sedimentation environments.

A hypothetical pathway of the process is shown in Figure 10. After binding of ions present in the solution by viruses, a small amount of $S^{2-}$ ions added to the mixture caused precipitation of fine structures due to the net surface charge close to 0 mV. Under such conditions, phenomena such as aggregation or agglomeration may occur. In experiments without viruses, the

charge is significantly more negative and thus, the colloid is stable (it does not precipitate immediately), so that no aggregation or agglomeration occurs at first. Subsequent addition of $S^{2-}$ ions releases a cascade process and the formation of larger structures.

There are several studies about the formation of framboidal structures built of different minerals. Kerridge described magnetite (Kerridge, 1970), Nuhfer and Pavlovic reported greigite (Nuhfer and Pavlovic, 1979), and Taylor characterized magnesioferrite

(Taylor, 1982). However, there are no reports on the formation of structures built of CuS and framboidal-like structures built of this mineral. Our experiments also failed to obtain CuS-framboidal-like structures.

## 5 Conclusions

We showed that viruses can potentially influence the precipitation of FeS and CuS crystals. However, it cannot be certainly

stated that the framboid structures were formed in our studies. In experiments with FeS, only framboid-like or spherical mineral structures have been obtained. What is more, these structures were found only in experiments conducted under static conditions without mixing, so it is possible that mixing interfered with the formation of framboid-like or spherical structures.  In CuS experiments, neither spherical nor framboid-like structures were observed. However, in both FeS and CuS experiments, the bacteriophages promoted the formation of finer crystallites compared to the control and influenced the agglomeration

processes, as well as changes in the net surface charge of the mineral-virus complex. The ζ-potential measurements have also revealed that the bacteriophages probably bind metal or sulphide ions, probably depending on the presence of a lipid envelope. Hence, considering our experimental results, it is possible that the presence of viral particles introduces one more factor that might influence the formation of framboidal structures. Our studies are preliminary investigations, which mainly consider the shape and physicochemical properties of precipitated minerals in the presence of viruses. Subsequent studies might include

the use of synchrotron techniques to show possible defects in the crystallographic structure caused by bacteriophages and, more importantly, to discover the time-dependent processes at the beginning of the crystallisation.

**Author Contributions:** P.D. and AB designed and conducted the experiments, analysed the data, and wrote the manuscript; M.D.S. performed the XRD and SEM analyses; K.K. performed the ζ-potential measurements; M.S. and Ł.Z. contributed to

writing the article.

**Competing interests:** The authors declare that they have no conflict of interest.

**Acknowledgments:** This study was supported by National Science Centre, Poland (OPUS 18, 2019/35/B/ST10/00719). We thank Grzegorz Kaproń from University of Warsaw for conducting XRD analyses, and dr. Mariette Wolthers from Utrecht University for discussion.

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

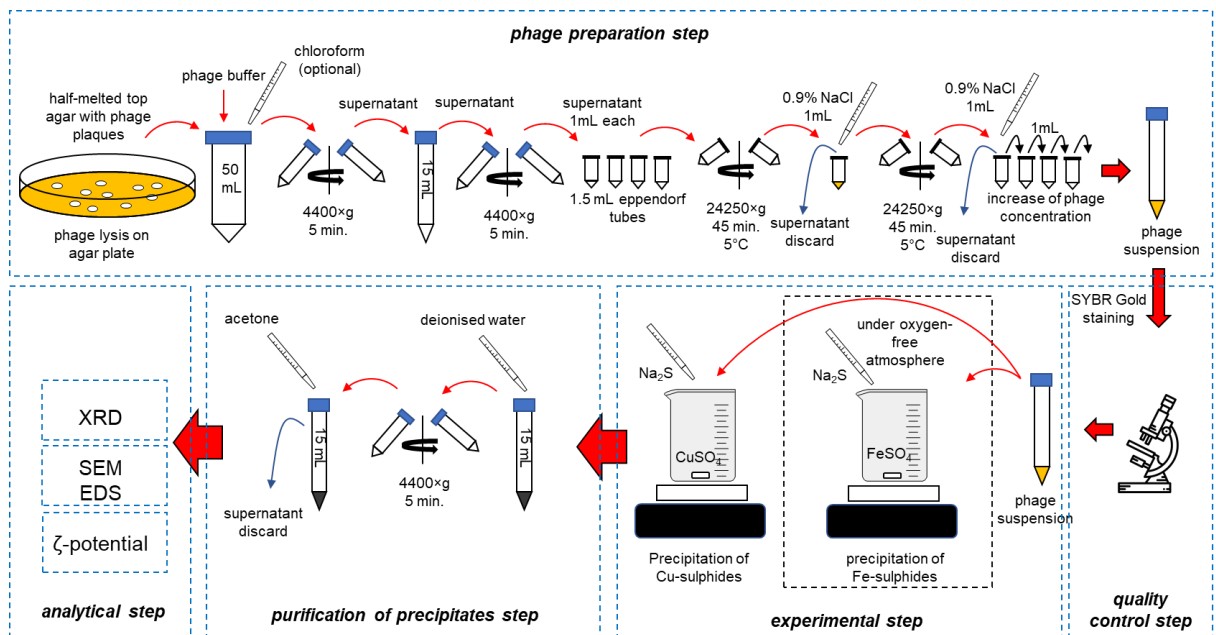

**Figure 1. The scheme of experiments.**


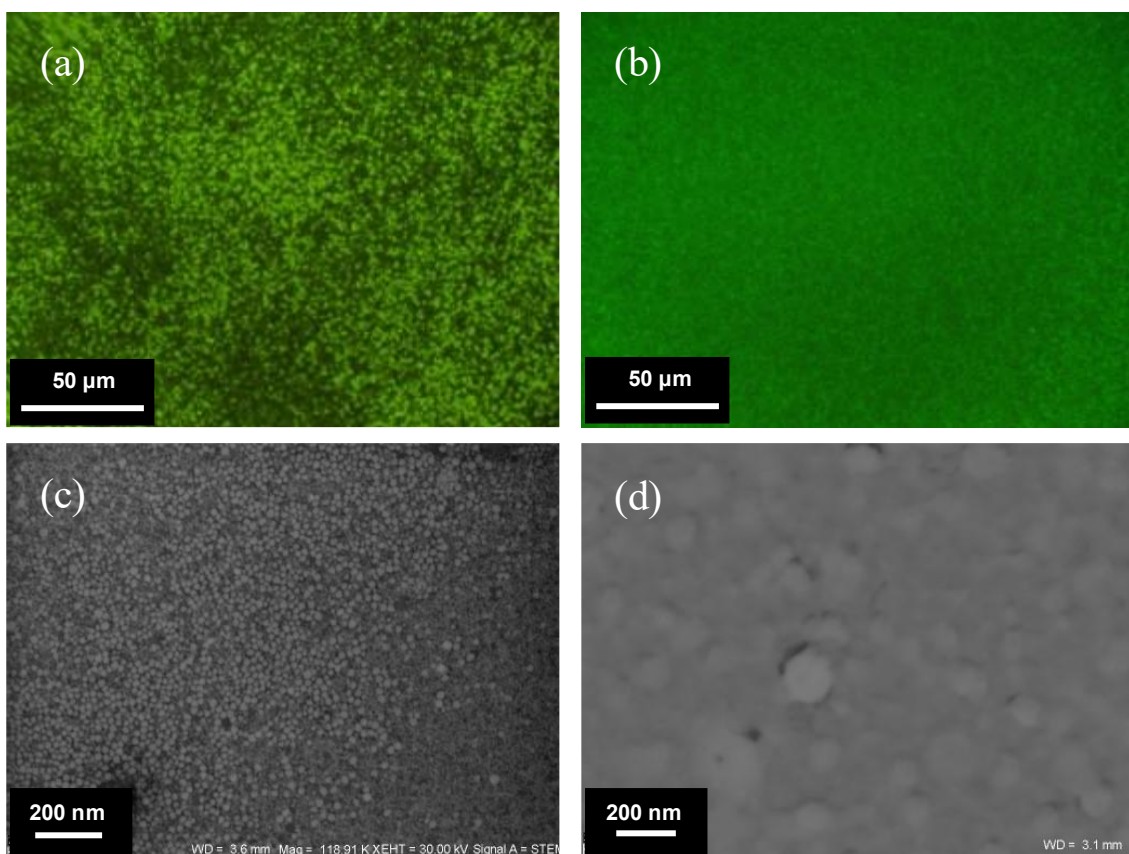

**Figure 2. Epifluorescence microscope images of purified bacteriophages stained with SybrGold®, (a)** *Escherichia phage P1*, **(b)** *Pseudomonas phage Φ6***; and SEM images, (c)** *Escherichia phage P1* **stained with uranyl nitrate, (d)** *Pseudomonas phage Φ6* **without staining.**

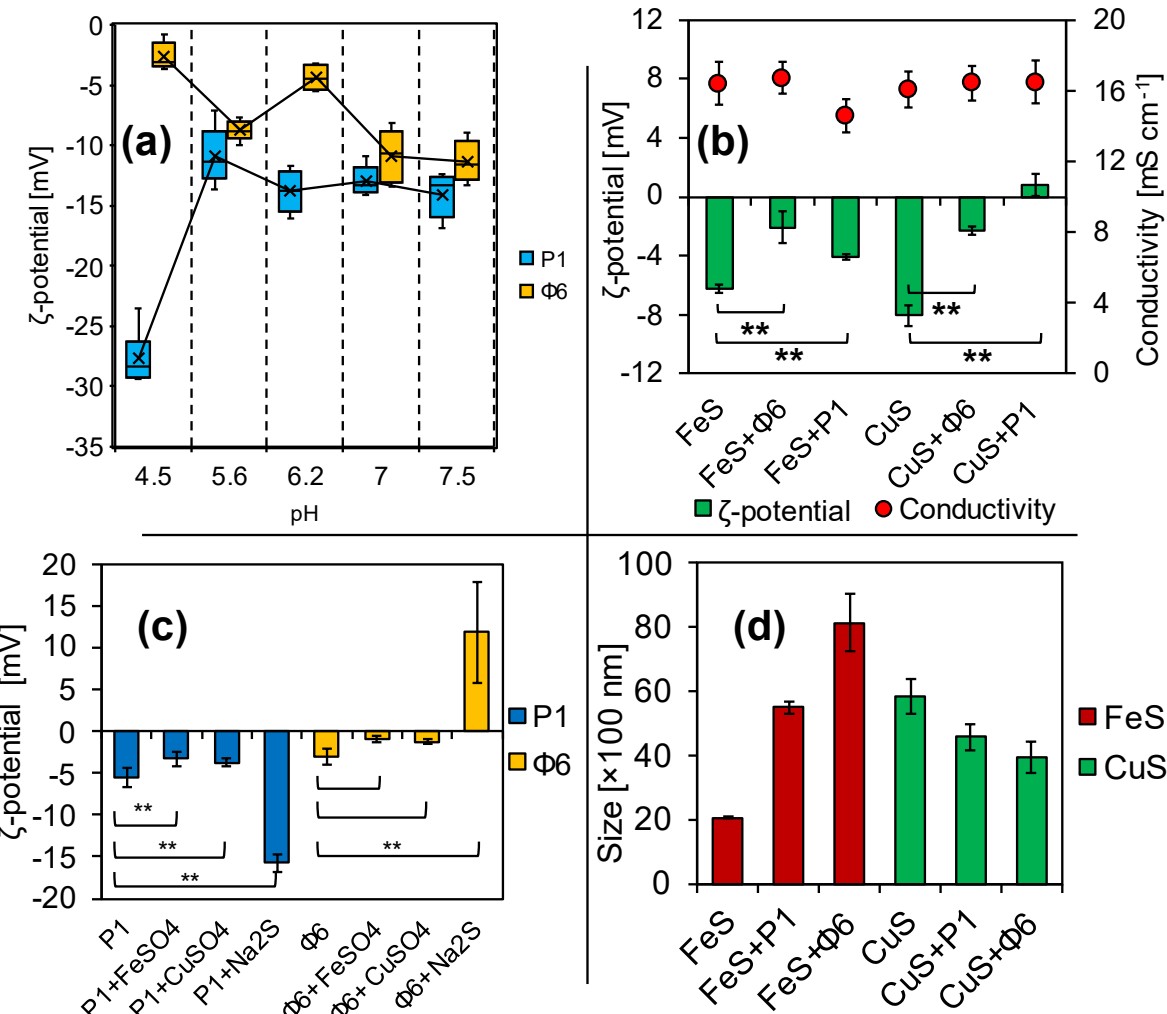

**Figure 3. Electrochemical properties of (a) viruses in phosphate buffers, (b) ζ-potential and conductivity of FeS and CuS precipitated with/without viruses directly after the experiment; (c) changes of the ζ-potential of bacteriophages with added solutions; (d) Z-average size distribution of FeS and CuS precipitated with/without viruses directly after the experiment. ** indicates statistical significance.**

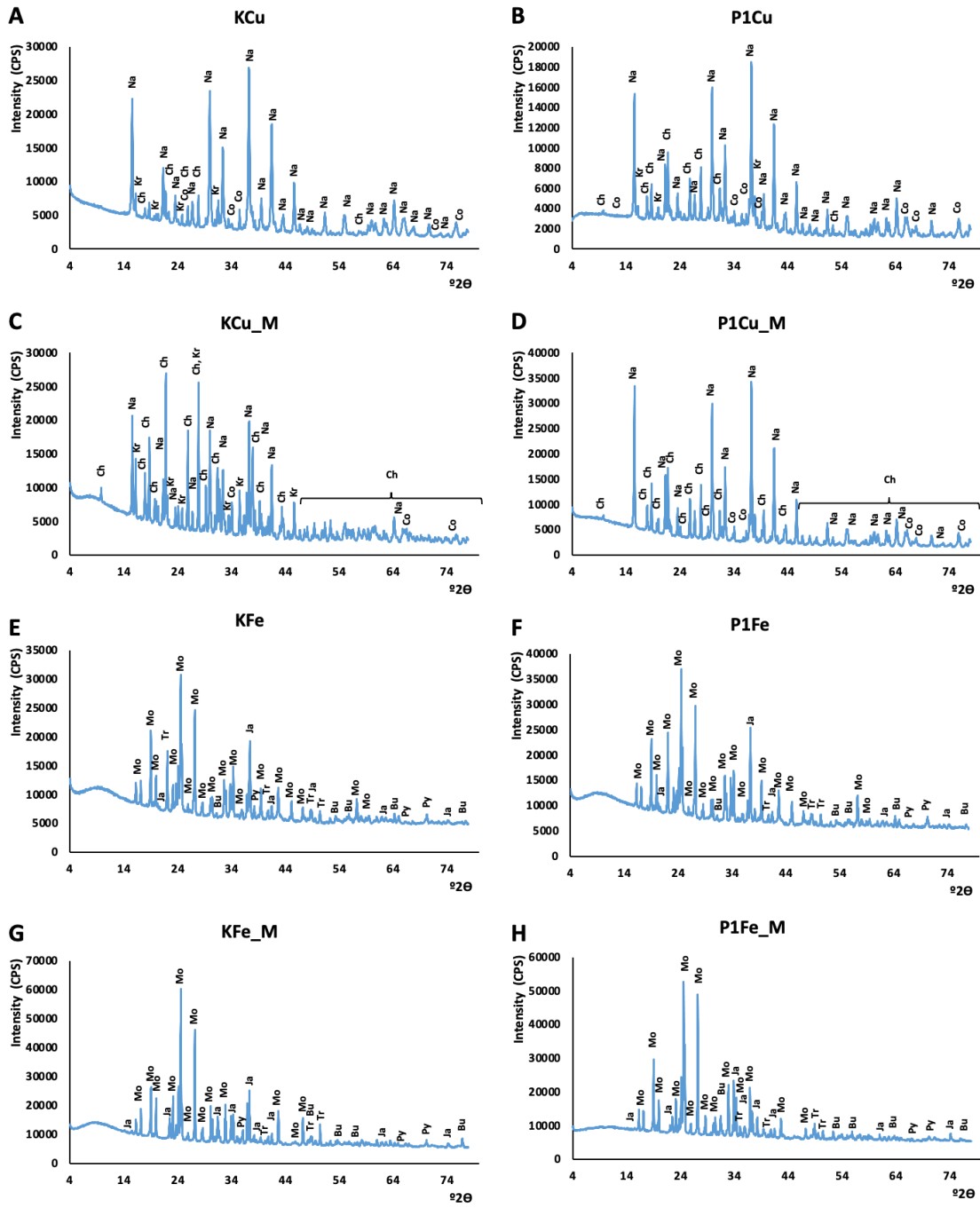

**Figure 4. XRD patterns of synthesized minerals. The CuS experiment without mixing: A – control experiment; B – P1 phage experiment. The CuS experiment with mixing: C – control experiment, D – P1 phage experiment. The FeS experiment without mixing: E – control experiment, F – P1 phage experiment. The FeS experiment with mixing: G – control experiment, H – P1 phage experiment. Oxidation products: Bu – butlerite, Ch – chalcanthite, Kr – kröhnkite, Na – natrochalcite, Ja – jarosite; substrates: Mo – mohrite; Sulphides: Co – covellite, Tr – troilite (?), Py – pyrite.**


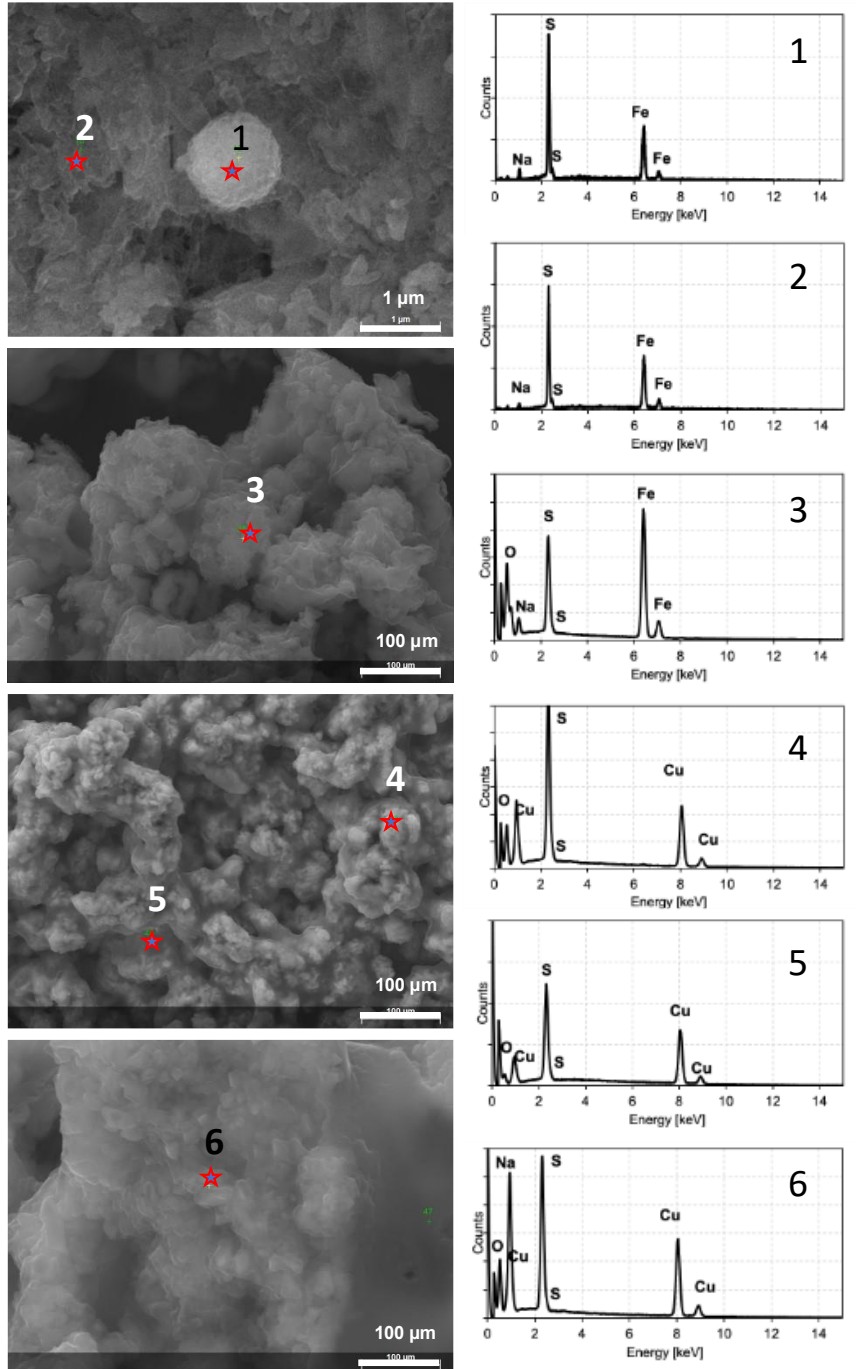

**Figure 5. SEM images and EDS spectra characterizing the observed minerals after experiments. Spectra 1 and 2 indicate FeS precipitates. Spectrum 3 probably indicates sulphates due to the strong oxygen signal in relation to sulphur. Spectra 4 and 5 indicate CuS minerals. Spectrum 6 indicates possible impurities due to the presence of sodium (strong signal in relation to sulphur).**

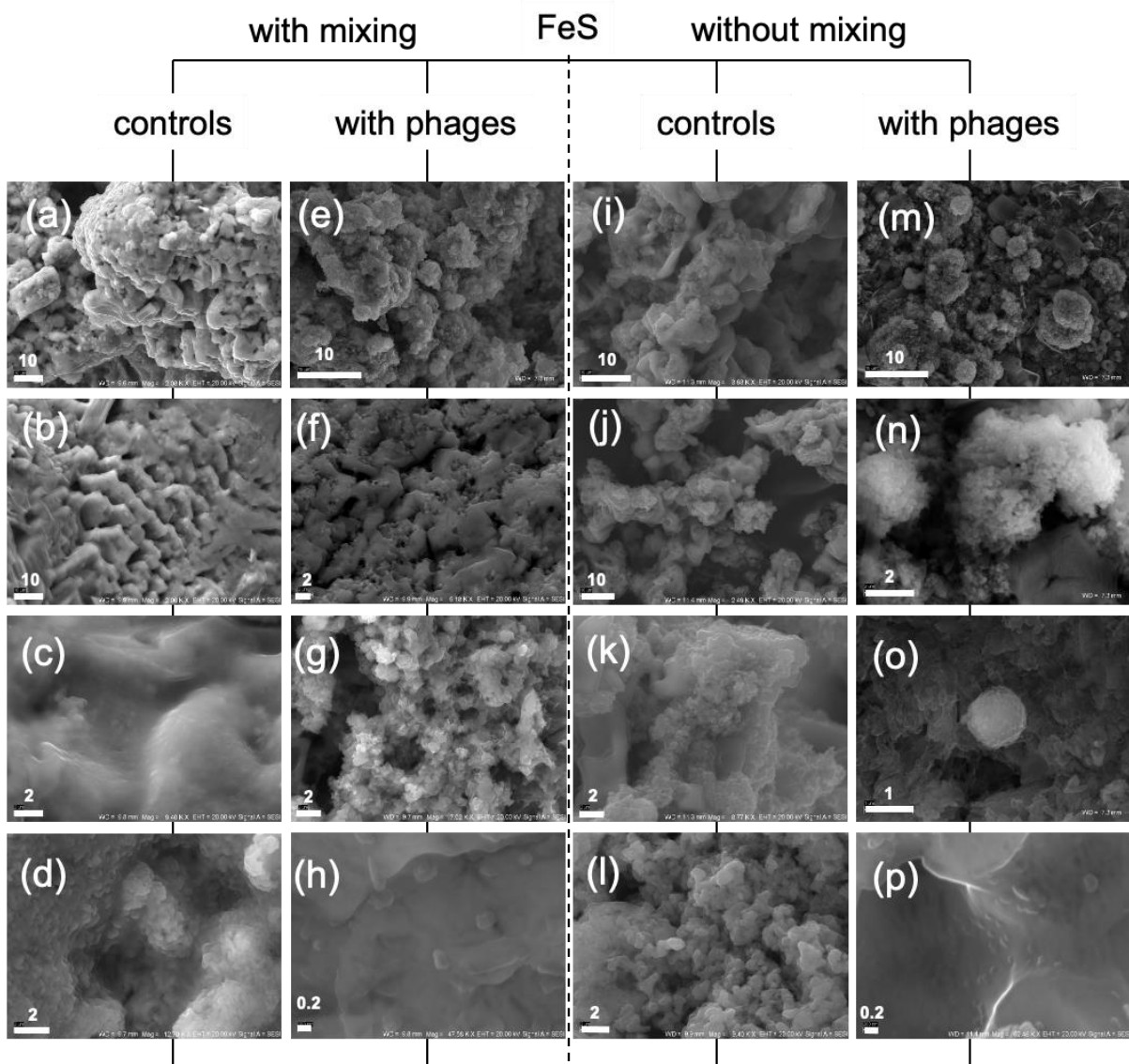


**Figure 6. FeS precipitates under SEM after experiments with and without P1 bacteriophages. (a-d) – minerals precipitated without bacteriophages under mixing conditions; (e-h) – mineral structures obtained in experiment with bacteriophages under mixing conditions; (i-l) – minerals precipitated without bacteriophages under static (without mixing) conditions; (m-p) – minerals precipitated with bacteriophages under static conditions; (m) and (o) show spherical structures resembling protoframboids. Bar**
**indicates scale in μm.**

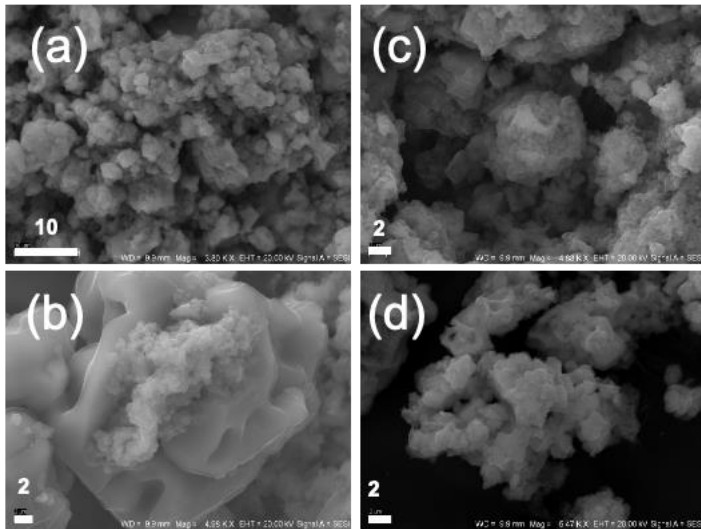

**Figure 7. FeS precipitates under SEM after experiments with Φ6 phage. (a-b) – minerals obtained in experiment under mixing conditions; (c-d) – minerals precipitated under static (without mixing) conditions. The bar indicates scale in μm.**


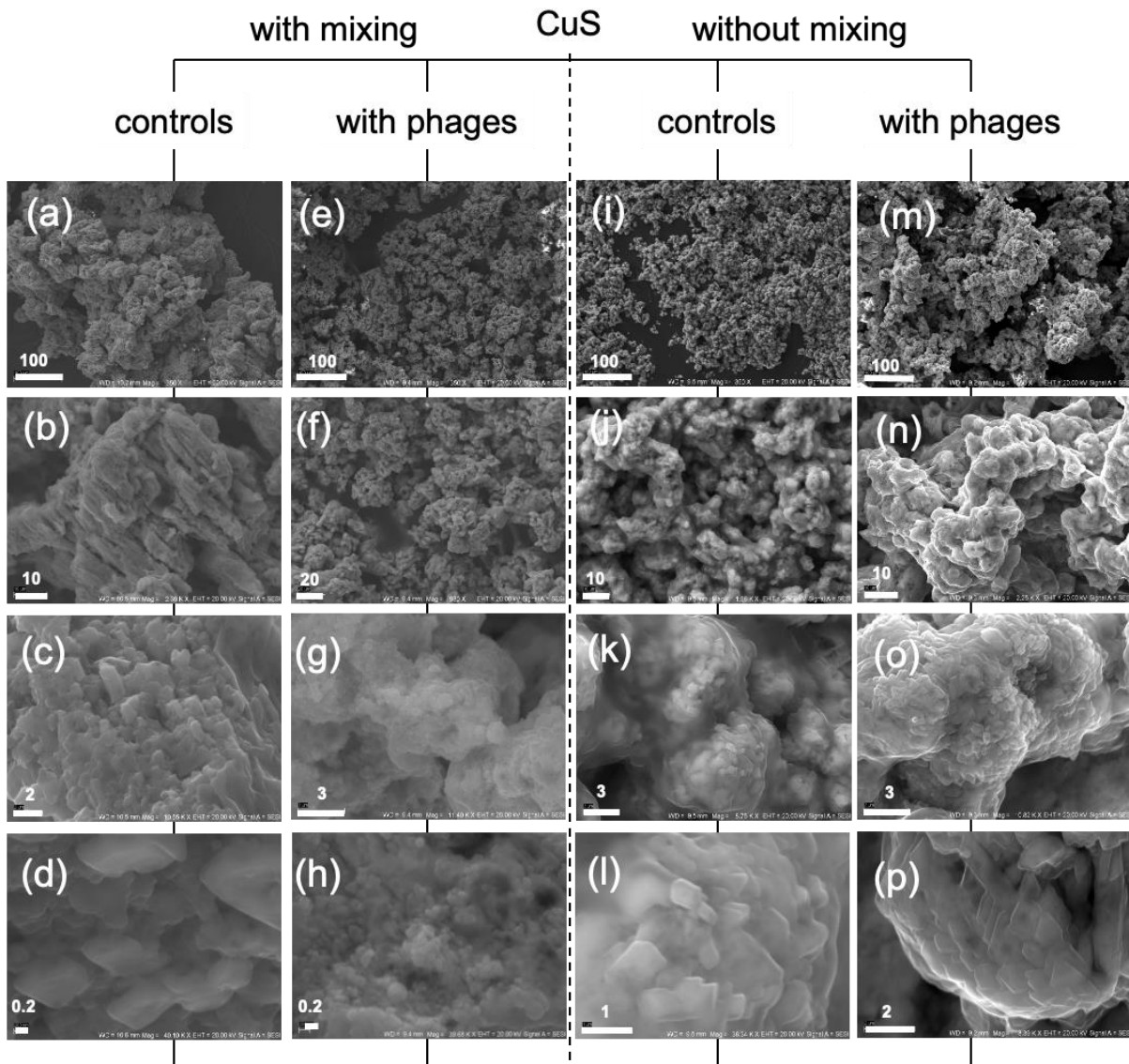

**Figure 8. CuS minerals under SEM after experiments with and without P1 bacteriophage. (a-d) – mineral structures precipitated without bacteriophages under mixing conditions; (e-h) – minerals obtained in experiment with bacteriophages under mixing conditions; (i-l) – minerals precipitated without bacteriophages under static (without mixing) conditions; (m-p) – minerals precipitated with bacteriophages under static conditions. Bar indicates scale in μm.**


Reprinted (adapted) with permission (Wolthers et al., 2005).

This study

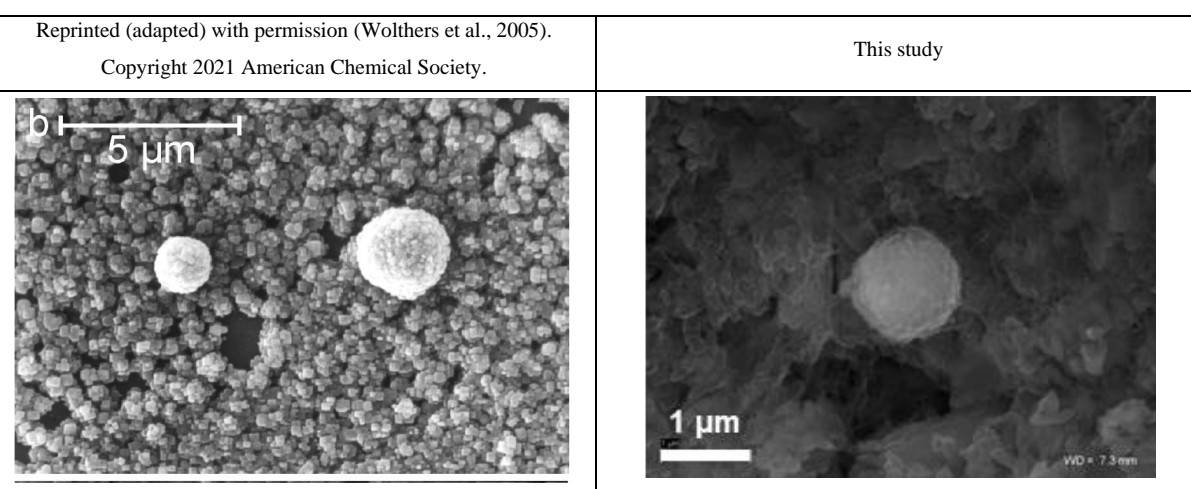

**Figure 9. The framboidal structures obtained by Wolthers (Wolthers et al., 2005) and obtained during our experiments. Note nearly identical shapes in both cases but under different experimental conditions.**

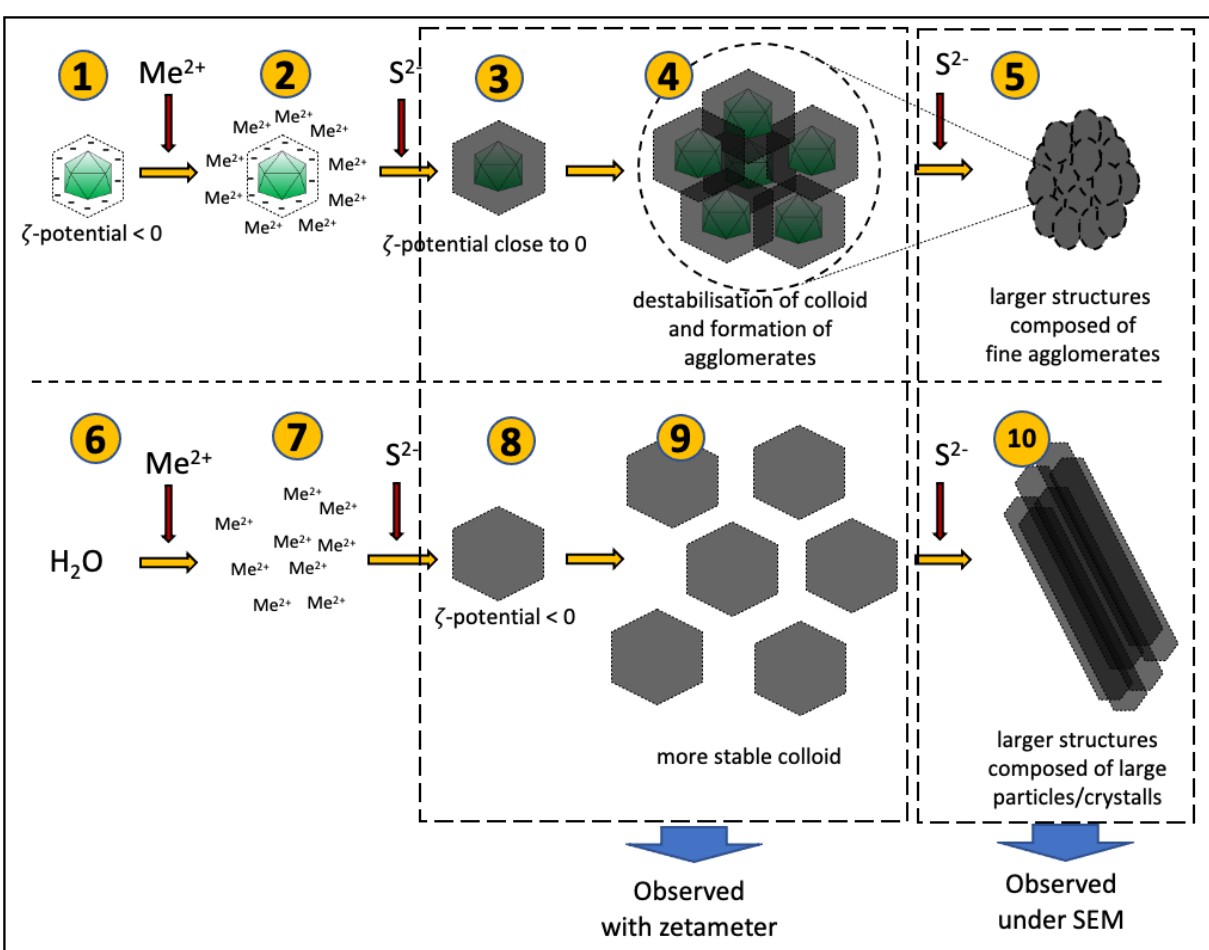

**Figure 10. Hypothetical processes leading to the observed phenomena.** Negatively charged bacteriophage capsids bind metal ions (1-2). When sulphide ions in small amounts are added, sulphides will precipitate on capsids (3). Due to surface charge close to zero, the colloid undergoes destabilization and agglomerates can be formed (4). Addition of more sulphides can promote creation of more such agglomerates which can lead to the larger structure (composed of small agglomerates) (5). The addition of small portion of sulphides in experiment without bacteriophages (6-8) can lead to the formation of more stable colloid due to lower $\zeta$-potential (9). However, the addition of more sulphide ions causes further growth of crystals and formation of larger structures composed of large particles or crystals.