# Peer review of "Do bacterial viruses affect the framboid-like mineral formation?"

_Biogeosciences, 2022_

## Author Comment (AC1)

We would like to thank the reviewers for their insightful discussion. We would like to emphasize that our experiments are the first attempt to determine whether bacterial viruses affect the formation of sulfide minerals. We cannot conclude the influence of bacterial viruses as there are not much data on this subject. But we believe that our preliminary research will give a rise to discussion and further, more in-depth studies. It is not easy to plan and perform such tests, for example, due to the difficulties with thorough cleaning of the viruses.  However, we know from other experimental studies that bacterial viruses (phages), so numerous in the environment, can strongly influence e.g., the formation of carbonates (see Słowakiewicz et al, 2021, *Geochimica et Cosmochimica Acta*). The precipitation of carbonate phases in the presence of phages revealed effects visible with the naked eye (strong agglomeration of crystallites as well as changes in their size, and formation of vaterite). We wondered if similar effects could occur in the case of sulfides. The first experiments with copper sulfides showed that it was! However, this topic is much more experimentally demanding. In our work, we show how we did it and how we cleaned our phages. Hence, bacteriophages could be used for research and experiments performed by other scientists.

We found that the reviewers' comments immensely improved our manuscript. Once again, we thank the reviewers. We will place all comments in the final version of the manuscript, referring to the suggestions of the reviewers. Our answers are as follows.

**Reviewer 1**

| Comment | Anwser |
|---|---|
| The Introduction is one long paragraph. It is unreadable as a coherent introduction. Consider breaking this text up. | We have rearranged the structure of the introduction. Now it consists of three main paragraphs. |
| Line 31, framboids are present in sulphides and oxides. | The sentence has been improved to be clearer. |
| Line34, spelling on euxinic | The spelling has been corrected. |
| Line 35, the text here is not accurate. Pyrite framboids are markers of the redox transition between oxygen-containing and anaerobic/sulphidic waters. In fact, some pulse of an oxidant is needed for pyrite formation since its sulfur atoms are present in the -1-valence state and precursor minerals like FeS are in the -2-valence state. | Thank you for clarifying this issue. We will complete this in the final version of the manuscript. |
| It is true that the Ohfuji paper looked at abiotic synthesis, but so did many other studies (Farrand et al. 1970; Berner 1969 Econ Geol v. 64; Graham and Ohmoto 1994 Geochim Cosmochim v. 58; Sweeney and Kaplan 1973 Econ Geol; Wilkin and Barnes 1996 Geochim. Cosmochim v. 60). These should be referenced, at a minimum, along with others. Seems to me that a very | Thank you for providing additional references. We have included them in the manuscript. |

| | |
|---|---|
| accurate and trustworthy evaluation of abiotic syntheses is required here as a point of comparison. | |
| Line 42, why Interestingly? | We think there was a misuse of the word. The sentence was rewritten. |
| Line 53, why should it be assumed? The Introduction is very fuzzy about the sizes and dimensions of bacterial viruses. The 50 to 200 nm size range makes to connection to any dimension in framboids that is consistent across the environments of their formation. And the image shown in Figure 2 is not connective to framboid sizes or shapes. The whole set up seems to be a stretch. | We have rearranged the paragraph and added an additional paragraph that includes more information about viruses and their involvement in ecosystems, as well as the behaviour.

Please note that Figure 2 shows the stained bacteriophages and not the structures obtained during this study. |
| Framboids are quite notably present in hydrothermal deposits yet this fact is quite conveniently omitted. Hydrothermal occurrence should be noted as a fact that may be contrary to bacteriophage involvement. | We have included additional references.

We can agree that hydrothermal occurrence of framboidal pyrite may stand against the bacteriophage involvement, but bacteriophages (or Archeoviruses) may also be present in such environments (up to 120°C). We have also included additional references related to this topic. However, the issue is still being investigated. |
| The experimental setup lacks any connection to natural settings and does not recognize the established understanding of pyrite formation. 1) describe redox and pH control; 2) describe aging. The use of the culture medium is an experimental requirement, perhaps. But it should be acknowledged that you are setting up the experiment to favor biology and not reproduce anything that happens in nature. | We are aware that the experimental setup is far from natural conditions. However, since the study is a preliminary project, we focuses on the setup where most conditions can be measured or standardized. An extension of the study is required, e.g., ionic strength or Eh were not controlled in this approach. |
| I was left unsure why so much emphasis was placed on the zeta potential experiments & data. Provide some introduction as to why this is important. | We included information about $\zeta$-potential and the importance of the measurements. |
| Results: the XRD data show that the experiments produced nothing that is realistic for natural settings. The traces of pyrite in the XRD patterns are not especially characteristic or convincing. Troilite is a high temperature phase; there should be no expectation for troilite identification. How were the samples collected for XRD studies? Please describe the controls that were in place to prevent oxidation. | We have added some explanations to the methodology in this regard. To avoid the oxidation of the samples, a glovebox with $N_2$ atmosphere was used. Moreover, the investigated samples were packed into the capillaries. The lack of characteristic pick of pyrite in XRD patters may be induced by sample oxidation and masked by substrates used for the synthesis experiments. Yes, troilite is a high temperature phase, and we do not have any information how the viruses may indicate the crystallization of this phase due to this is a preliminary study or it is miss identification of this phase. We know that troilite is unlikely to form in the experimental setup. However, in quite noisy diffraction patterns, the identification of individual phases may be erroneous. We decided that if the adopted |

| | identification algorithm shows such a phase, we place it on the diffractogram. Due to our fault, there was no additional comment that should be included in the manuscript, as suggested by the reviewer. In addition, research on this type of material is very troublesome. Sulfide minerals precipitated in the presence of biological material such as bacteria or phages seem to be extremely susceptible to oxidation. Despite the use of a nitrogen-filled chamber and XRD capillaries, we did not avoid the formation of oxidation products. |
|---|---|
| In the SEM studies, did you use backscatter detection? This would have revealed the high atomic Z particles more precisely. | During this study we used the InLens and BSE detector. |
| The images shown in Figure 6 look less like framboids then the kinds of precipitates that form in abiotic experiments. There is no improvement in morphological connection to framboid structures, in fact a step back, which leaves the reader unconvinced that this represents anything new. The weak similarity should be acknowledged. | Our research is certainly inconclusive and uncertain whether phages can influence the formation of framboids. However, we clearly show that pages influence e.g., the size of the mineral phases formed in their presence. This effect is very clearly visible with the naked eye during experiments, which we pay attention to especially during the precipitation of copper sulphides. In addition, phages affect the degree of agglomeration of fine crystallites, possibly favouring the formation of spherical forms. We do not claim, however, that the structures we obtained are framboids, hence we always use the phrase "framboid-like". We give rise to further discussion on this topic. This is only the first such experimental study to discuss the role of phages in the formation of sulphide minerals at all. |
| Line 270, microcrystals in framboids come in multiple shapes and sizes. The link to Ohfuji and Akai is a weak thread. Please acknowledge the full range of microcrystal sizes and shapes. | We have rearranged the paragraph to make it clearer. We have also included an extra table with more information about sizes (please find Table 1 attached at the end of the document). The information about the shape of microcrystals has been also included in the manuscript. |
| Line 277, pyrite framboid size in nature depends on the environment of formation (it is not based on the study - not sure what this means). A coherent model would need to explain why viruses would produce different sized framboids in different environments. | |
| Line 284, again there are many abiotic framboid synthesis examples, all of them produce even more framboid-like morphology than what is described here (see references above in point 5; please consider adding these to the discussion for an objective analysis). | We have rearranged the whole paragraph and included a table with an overview of experimental conditions (please find Table 2 attached at the end of the document). We have also included the publications mentioned in Point 5. |

**Reviewer 2**

| Comments | Answers |
|---|---|
| Lines 14-16: Please re-write or delete this phrase. In the context of biomineralization, ions are not attracted by biological surfaces – or are capsids electrically charged, or magnetic? Please explain how could a virus change the electrochemical properties of precipitated minerals. | We have deleted the misleading part of the phrase. We think that the net surface charge of the minerals precipitated with viruses can be changed. We included it in the discussed section. |
| Line 18: They are different viruses, and thus it was expected different outcomes in the experiments. The presence of a lipid envelope is certainly a difference between the two viruses, but note that cells have membranes and are frequently involved in biomineralization. | We have rearranged the sentence. We are aware cells have lipid membranes that can be involved in biomineralization. We just wanted to point out that the lack of a lipid membrane in the case of phages might have an impact on the formation of framboid-like structures, but we do not claim that this factor is crucial. |
| Lines 21-22: (i) see comments for lines 14-16; (iii) what colloid? The structure of framboids indicate they form as a sphere of crystallites, not by agglomeration. | We assumed that viruses could facilitate the formation of tiny crystallites, that could form more complex forms at a later stage. We do not state that framboids arise from agglomeration. We argue that viruses, e.g., by changing the net surface charge, can favour the formation of very fine crystallites. This process can be seen with the naked eye - the presence of viruses causes the formation of very fine structures (this fact it is even used in the synthesis of nanostructures). In contrast, virus-free synthesis produces a massive precipitate. We have changed the description of the process. |
| Lines 28-29: Microbes can oxidize both sulfide and Fe(II) from pyrite and other sulfide minerals, producing sulfate, Fe(III) minerals and protons, leading to decrease in pH. | We have supplemented the sentence with additional information. |
| Lines 60-65: Please re-write these sentences. The hypothesis/objectives are not clear. It is unclear what kind of experiments were done.

In addition, the importance of your findings is over-emphasized – data presented are not sufficient to access the importance of the formation of framboids associated to viruses in nature. | We have rewritten the paragraph. Please note, that we do not state that the structures obtained are framboids, but framboid-like. We also suggest that bacteriophages might be a factor in one of the ways of framboid-like structures formation. |
| Lines 111-117: Were the solutions treated to remove dissolved oxygen? How? Was O2 in the solutions quantified? | We have included the information. It was omitted by our mistake. The concentration of oxygen in the glovebox was controlled with an oxygen meter. The water prior to use was degassed by autoclaving (121°C, 15 min). |
| Lines 127-128: ions are not attracted by viruses: they adsorb, bind, etc. | We changed the misleading word to "adsorb" in the whole manuscript. |
| Line 130: Were the viral capsids isolated? How? How binding of Fe, Cu and sulfide to viruses were measured? | The viral capsids were not isolated. The whole solution was placed in the measurement cell. The apparatus was used to measure the $\zeta$-potential of viruses with adsorbed cations or |

| | anions (change in charge of electrical double layers). |
|---|---|
| | A sentence describing the process was added. "Subsequently, the solutions were transferred to measurement cells and measured." |
| Lines 139-143: Please explain how samples were prepared for XRD. Of crucial importance is how samples were stored, and how long they were stored. | We have included additional information. We put great effort to prevent the oxidation of the samples. However, the transfer of the samples to the capillaries caused partial oxidation. We pointed out this problem in the results. |
| Line 141: there is no "CoKα lamp" in X-ray diffractometers. | We have changed "CoKα lamp" to "CoKα radiation source". |
| Line 148: Are you sure it was a glass slide? Glass coverslips are widely used to mount samples for SEM. | We use glass slides due to their better mechanical properties. Our protocol was adjusted for glass slides. Glass coverslips are very fragile and may break during the procedure. |
| Line 148: Are you sure it was 20 nm? Or was this an estimate from the device? | The layer was 20 nm thick, and it was applied by a device. |
| Lines 170-171: …are differences in the samples with and without viruses (or bacteriophages). Were the differences noted in the z-potential or the conductivity? | We changed the sentence to be clearer. The significance test was performed among the same group of metals. |
| Lines 171-172: were the significant differences found between the FeS and CuS groups, or among the three treatments for the same metal? | |
| Lines 177-178: Was the z-potential in the last column measured with virus + metal + Na2S, or with virus + Na2S? | We have clarified the sentence. The $Na_2S$ (similarly to $FeSO_4$ or $CuSO_4$) was added as sole solution. |
| Lines 188-194: Any differences in the mineral composition were probably masked by sample oxidation. This is stated in lines 193-194. But the sentence in line 188 states that "Phase composition of samples did not differ significantly". You could re-arrange the text to make clear in the beginning of the paragraph that the X-ray data should be analyzed knowing that the sample has been oxidized. | Mohrite, Butlerite and jarosite are the product of oxidation. The list of the phases which are results of oxidation of the sample was added to the text. Substrates: mohrite Oxidation product: butlerite, chalkantite, kröhnkite, natrochalcite, jarosite Sulfates: troilite, pyrite |
| Are mohrite, blutlerite and jarosite products of oxidation of the sample? You should make a list of probable original products and oxidation products for the FeS and CuS experiment, to help the reader unfamiliar with these minerals. | |
| Lines 197-198: How did you recognize the mineral phases in the SEM to record the EDS spectra? Maybe the use of the term "mineral phase" is a mistake. Do you mean "minerals", "mineral particles", or something like that? | There was a misuse of the term "mineral phases". We have changed the term to 'minerals' in the entire manuscript. |
| Line 200: the use of "mineral phases" is probably a mistake here too. | |

| | |
|---|---|
| Lines 200-201: The spectra "1" and "2" seem very similar, not distinct. | We changed the sentence to: "Spectra 1 and 2 revealed clearly visible signals from iron and sulphur." |
| Line 204: O and Na are not negligible in spectrum 4, and Na is not negligible in spectrum 5. | The presence of O, and Na in spectrum 4, and Na in spectrum 5 is not negligible. However, the analytical line of Na Kα is overlapping with the Cu Lı line so precise estimation of how much Na is in the sample is very hard even if the deconvolution of the analytical lines is used. Moreover, it is very hard to tell if the presence of Na in the investigated crystallites due to the electron beam may also excite phases below the cooper sulfate crystallites. |
| Lines 209-216: The text is very confusing. From the images in Figure 6, it seems that the shapes and sizes of the particles are not really different in the three first groups of images (a-l). In figures m-p they seem smaller, and sometimes are arranged in spheres. | We would like to apologize if we were not very clear. We wanted to show that phages cause tiny crystallites to aggregate into spherical structures frequently. In addition, in experiments with phages, many fine 200-300 nm crystallites are formed, which are absent in abiotic experiments. We indicate that the differences are not significant, but quite pronounced, especially with the P1 phage and with mixing vs. no mixing. Spherical structures never arise in mixing experiments, not even with phages. In the case of the experiment with Phi6 phage (Fig. 7) we note in the text that there are practically no visible differences here. Only an overrepresentation of very fine minerals can be seen, similar to P1. |
| Line 210: "Experiments with bacteriophages gave similar structures". Similar to what? To each other? | |
| Lines 222-223: From the images, images in Figure 7 not seem to be really different from images in Figure 6 without bacteriophages. | |
| Lines 234-235: The differences observed with the naked eye must be diverse from that observed at the SEM – the magnification are orders of magnitude different. | |
| Line 242: Please provide a reference for the DLVO theory. | We have also included the reference regarding DLVO theory. |
| Lines 242-243: Again: viruses are not nuclei which attract ions from solution. If they attracted ions from solution, they would be soon encased in minerals and would not be effective in infecting cells (which are the sine-qua-non condition for virus replication). | We have changed the misused word to "bind". Kindly note, that we are aware that viruses can infect bacterial cells only under proper conditions, when capsids are not mineralized. But what is important, even when the infection takes place, bivalent ions (like $Mg^{2+}$) are needed. However, in this research we did not study the process of infection when the saturation index is exceeded. |
| Line 244: the work cited states that capsids can bind iron and nucleate iron minerals. | As we mentioned, the misleading word "adsorb" has been changed in the whole manuscript. |
| Line 249 Viruses do not attract ions; they can bind, or adsorb, ions. | |
| Lines 251-252: bind, not attract. If they bind, you can measure differences relatively to the sample without Na2S. | |
| Lines 254-255: It is probably much more complex than just charge interactions. For example, sulfide ions may interact with -SH groups of cysteine residues. | We are aware that there can be many other phenomena.  We have included a hedged statement. |
| Line 261: Are the two groups statistically different from each other? Or each experiment is different from the others in the same group? | The statistical significance test was performed among the same metal group. We have rewritten the sentence to remove the ambiguity. |

| | |
|---|---|
| Lines 268-268: The measurements are not erroneous if the effects of aggregation inherent to the technique are considered in interpretation. Even if the numerical results do not represent real sizes, they can still be used to compare samples. | Yes, we agree with your comment. However, we wanted to note, that the results may be somehow erroneous. |
| 271-273: Was the P1 virus chosen because of the icosahedral shape, or the icosahedral shape was a coincidence? Several minerals can show icosahedral morphologies, including macroscopic specimens of natural pyrites. Crystal shapes depend mainly on the arrangement of atoms, and this is the case of icosahedral pyrites. Conversely, there are several icosahedral viruses. The fact that pyrite in framboids and in your experiments is icosahedral does not mean that they were nucleated by icosahedral viruses, since they can be produced by purely chemical processes. | We do not state that the icosahedral shape of the virus is a *sine qua non* for framboidal pyrite formation. The P1 virus was chosen as an example of a common family of "crystalline" shaped viruses. The Phi6 virus was selected as an example of a lipid enveloped virus. We did not know in advance what effect such a difference could have. Most likely, the two viruses have different effects on mineral precipitation, possibly due to the presence or absence of a lipid envelope. We indicate this effect, but it requires more careful study. |
| Lines 300-302: The idea or virus capsids attracting ions again – please consider that binding is the important thing to consider for mineral precipitation. To precipitate a mineral, it is needed several layers of each type of ion. | We have changed the misused word in the whole manuscript. We have also considered all your comments on this issue.

We are not sure about the influence of the lipid envelope. Therefore, we only assumed that the lipid envelope is a differentiating factor. We aimed to check this phenomenon, and thus we chose two morphologically different bacteriophages in the preliminary studies. |
| Lines 318-319: Again the idea of viruses attracting ions from solution. Here, attract and bind seem to have been used as synonyms. Are you sure that the lipid envelope is the cause of the differences observed between the viruses? They surely have other differences too. | |
| Lines 322-323: Fe(II) and S(II-) are not oxidized in anoxic environments, and FeS precipitation is not limited in these environments. Most of the Earth's crust is anoxic, including deep soils and sediments. | Thank you for your comment. We are aware of this problem. The ambiguous sentence will be changed in the final version of the manuscript. |
| Figure 2: Please provide scale bars for the light micrographs. | |
| Figure 2 (caption): Magnification in printed micrographs is almost useless – it depends on how much the image is enlarged. Please write the bacterial genus Pseudomonas in italics. Please state in the caption the technique used (fluorescence light microscopy or transmission electron microscopy) in each image. | The scale bars have been added. We have changed the captions. |
| Figure 3b: Please make a legend without colors for Figure 3b, to make it clear the differences between conductivity and z-potential in the graph. | This will be changed in the final version of the manuscript. |
| Figure 3 (caption): Please explain what where the solutions used for measurements (phosphate buffer, saline, etc) in each plot.

In (b), there is a plot of conductivity which is not mentioned in the caption.

In (c), it is shown the z-potential of complex materials in suspension, not the "attraction of ions | We have included additional information regarding the solution used, as well as information about conductivity. All the arisen issues will be changed in the final version of the manuscript.

(**) means statistical differences among datasets represented on bars in Figure b and c (mean value from 3 measurements). |

| | |
|---|---|
| by bacteriophages" – there are other ingredients in the mixture.

In (d), it is shown the size of FeS and CuS particles. Does ** means statistical differences? It is not clear what data were compared in statistical tests. | |
| Figure 4: Please highlight the peaks for the minerals which were not oxidized during sample preparation. | Thank you for your comment. This will be corrected in the final version of the manuscript. "Ja" is the description for jarosite. This was mistakenly omitted in the caption. |
| Figure 4 (caption): Please separate Fe and Cu minerals in the list at the end of the caption, and provide a chemical formula for each mineral. You can see mindat.org for mineral formulae. In the figure, is "Ja" for jarosite? | |
| Figure 5: The lettering in the scale bars are too small. Please increase them. | |
| Figure 5 (caption): How was this material prepared? With viruses, or not? | These are only sample images, and the EDS spectra were used to show how we tentatively differentiated the mineral structures under SEM. As is well known, mineral structures are not always easy to distinguish under SEM, but using EDS spectra it is easy to determine whether we are dealing with sulphides or with sulphates and other oxidation products. This is what EDS spectra served, and that is why we show them only as an example. |
| Figures 6 and 8 (captions): It is mentioned in the captions that the experiments were made with the P1 bacteriophage, but a-d and i-l were prepared without bacteriophages. | We are sorry. That was our mistake. This is an unclear caption under the figure. Of course, the experiment was performed with phages and without phages as a control. |
| Figure 10: The idea of icosahedral viruses serving as nuclei for the synthesis of icosahedral FeS mineral particles, expressed in the drawings and also in the text, is not based on your data, nor on data from the literature. It seems pure imagination. A second issue is that there are two possibilities for formation of spherical structures: agglomeration as suggested in Figure 10, or they are formed already as a sphere. For framboidal pyrite, it seems more likely that it forms as a sphere: otherwise, how could they be so well organized by aggregation of pre-existing particles? The "framboids" shown in other figures of this manuscript are too smooth to have been formed by aggregation of pre-existing particles. | We did not want to confuse the readers. If we were not clear enough, we would like to apologize. We do not claim anywhere directly that the icosahedral structure of the virus causes the formation of framboid pyrite. We are merely claiming that the "crystalline" structure of viruses such as P1, which happens to be icosahedral, but does not to be so, is likely to facilitate the formation of some minerals or their crystalline forms.
 The angular structure of the virus in this figure corresponds to the above-mentioned problem but does not indicate the icosahedral structure. The problem described here has been recently discussed e.g., in these papers:
Słowakiewicz, M., Borkowski, A., Syczewski, M. D., Perrota, I. D., Owczarek, F., Sikora, A., Detman, A., Perri, E., and Tucker, M. E.: Newly-discovered interactions between bacteriophages and the process of calcium carbonate precipitation, Geochim. Cosmochim. Acta, 292, 482–498, 2021
Perri, E., Tucker, M. E., Słowakiewicz, M., Whitaker, F., Bowen, L., and Perrotta, I. D.: Carbonate and silicate biomineralization in a |

| | hypersaline microbial mat (Mesaieed sabkha, Qatar): Roles of bacteria, extracellular polymeric substances and viruses, Sedimentology, 65, 1213–1245, 2018.
Perri, E., Słowakiewicz, M., Perrotta, I. D., and Tucker, M. E.: Biomineralization processes in modern calcareous tufa: Possible roles of viruses, vesicles and extracellular polymeric substances (Corvino Valley – Southern Italy), Sedimentology, 69, 399–422, 2022. |
|---|---|
| **Technical comments:** | |
| Lines 13 and 19: do you mean Enterobacter, or enterobacteria? If it is Enterobacter (genus), it should be written in italics; if it is enterobacteria (common word), it should be written without capital letter. | According to DSMZ, we changed the name to "*Escherichia phage P1*" in the whole manuscript.
https://www.dsmz.de/collection/catalogue/details/culture/DSM-5757 |
| Lines 14 and 19: Pseudomonas should be written in italics, since it is the name of a bacterial genus. | The name has been marked in italics. |
| Line 29: sulphide. | The spellings have been corrected. |
| Line 33: Please add a space before the parenthesis. Is it greigite? | |
| Line 34: euxinic? | |
| Line 40: I suggest to begin a new paragraph to explain the basics of viruses and bacteriophages. | The paragraph has been rewritten. Additional information about viruses has been added. We have included information about their *life* modes*, types of infections, and the impact on the environment. |
| Lines 42-43: the same idea is expressed better in lines 51-53. | |
| Line 49: I suggest to begin a new paragraph to present the phages used in this work. | |
| Lines 51-53: Consider using "can" only once. | The sentence has been rewritten. |
| Lines 60-61: I think "in this work" would be better. | "In this paper" has been replaced with "in this work" |
| Line 75: laminar flow cabinet? | "Laminar chamber" has been replaced with "laminar flow cabinet" |
| Line 89: You may use "used" instead of "added". | "Added" has been replaced with "used" |
| Line 137: the device. | The preposition has been included. |
| Line 164: "both" is not suitable here. | We have changed "both" for "studied". |
| Line 192: There is a misspelling here, please write "chalcanthite". | The misspelling has been corrected. |
| Figure 4, caption: There is some misspelling: synthesized, in the mineral names, please write "chalcanthite" and "troilite" (you can see mindat.org for correct mineral names). | We have checked the names on the provided website. The misspellings have been corrected. |
| Lines 197-198: EDS spectra are not measured, they are obtained. | We have changed "measured" to "obtained". |
| Lines 209-210: visibly small | The spelling has been corrected. |
| Line 244: viral | |
| Lines 287-288: the hydration of the FeSO4 is excessive detail. | We have removed excessive details. |
| Lines 295-297: Try writing a single phrase comparing your results with others'. | We have rewritten the sentences. |
| Lines 298-299: Try rewriting this phrase using "stir" only once. | |

Table 1.

| Microcrystal diameter [μm] | Framboid diameter [μm] | Source | Reference |
|---|---|---|---|
| 2-3 | - | Precambrian rocks with copper and lead-zinc ore; Mount Isa Shale | (Love and Zimmerman, 1961) |
| 0.12 | 12 | | |
| 0.9 | 10 | Deep-sea sediments; Angola Basin | (Schallreuter, 1984) |
| 0.7 | 12 | | |
| 2 | 24 | | |
| 0.3 – 0.7 | 3 - 10 | Super-anoxic fjord; South Norway | (Skei, 1988) |
| - | 1 - 2 | Coal basins; Bulgaria | (Kortenski and Kostova, 1996) |
| - | 50 - 70 | | |
| 1 | 10 - 15 | Mudstone; Lower Eocene, Marquez Shale | (Collins, 1982) |
| - | 30 - 80 | Muddy sediments (Miocene – Holocene) | (Ohfuji and Akai, 2002) |
| - | 5 - 20 | Modern reductive sediments | |
| 0.5 | 5 - 20 | Sulphur microbial mats; Kane Cave | (Folk, 2005) |
| 0.8 - 2 | 6 - 12.5 | Methane-derived carbonate chimneys; Gulf of Cadiz | (Merinero et al., 2009) |
| - | <200 | Sedimentary rocks of the gold deposits (Paleozoic); Nevada, Victoria, USA | (Scott et al., 2009) |
| 0.3 - 5 | 3 - 10 | Sediments in the South Caspian Basin | (Kozina et al., 2018) |

Table 2.

| Reagents | Temperature [°C] | Duration | Reference |
|---|---|---|---|
| $FeSO_4$, $H_2S$, $S^0$ | 65 | 2 weeks | (Berner, 1969) |
| $FeSO_4$, $H_2S$, $CaCO_3$; glycerine | 23 | Up to 1 year | (Farrand, 1970) |
| $FeCl_2$, $H_2S$, $S^0$ | 25, 60 or 85 | Up to 6 days | (Sweeney and Kaplan, 1973) |
| $FeCl_2$, $FeSO_4$, $Fe(NO_3)_3$, $Fe(NH_4)_2(SO_4)_2$, | 25; 100 | 2 days; 4 months | (Luther, 1991) |
| HCl, NaCl, FeS, CaSO4 | 150 - 300 | Up to 8 weeks | (Graham and Ohmoto, 1994) |
| Mackinawite or greigite, $H_2S$, | 70 | - | (Wilkin and Barnes, 1996) |
| $Na_2S$, $Na_2O_3Si$, $FeCl_2$, $Fe(NH_4)_2(SO_4)_2$, $Fe(NO_3)_3$ | 23 | Up to 2 years | (Wang and Morse, 1996) |
| FeS, $H_2S$, $KH_2PO_4$/$K_2HPO_4$; Ti(III) citrate | 60 - 100 | Up to 45 days | (Butler and Rickard, 2000) |

---

## Author Response (AR1)

We would like to thank the reviewers for their insightful discussion. We would like to emphasize that our experiments are the first attempt to determine whether bacterial viruses affect the formation of sulfide minerals. We cannot conclude the influence of bacterial viruses as there are not much data on this subject. But we believe that our preliminary research will give a rise to discussion and further, more in-depth studies. It is not easy to plan and perform such tests, for example, due to the difficulties with thorough cleaning of the viruses. However, we know from other experimental studies that bacterial viruses (phages), so numerous in the environment, can strongly influence e.g., the formation of carbonates (see Słowakiewicz et al, 2021, *Geochimica et Cosmochimica Acta*). The precipitation of carbonate phases in the presence of phages revealed effects visible with the naked eye (strong agglomeration of crystallites as well as changes in their size, and formation of vaterite). We wondered if similar effects could occur in the case of sulfides. The first experiments with copper sulfides showed that it was! However, this topic is much more experimentally demanding. In our work, we show how we did it and how we cleaned our phages. Hence, bacteriophages could be used for research and experiments performed by other scientists.

We found that the reviewers' comments immensely improved our manuscript. Once again, we thank the reviewers. We placed all comments in the final version of the manuscript, referring to the suggestions of the reviewers. Our answers are as follows.

| Comment                                                                                                                                                                                                                                                                                                                                                                                       | Anwser                                                                                        |  |  |  |
|-----------------------------------------------------------------------------------------------------------------------------------------------------------------------------------------------------------------------------------------------------------------------------------------------------------------------------------------------------------------------------------------------|-----------------------------------------------------------------------------------------------|--|--|--|
| The Introduction is one long paragraph. It is
unreadable as a coherent introduction. Consider                                                                                                                                                                                                                                                                                              | We have rearranged the structure of the introduction. Now it consists of three main           |  |  |  |
| bleaking this text up.                                                                                                                                                                                                                                                                                                                                                                        | paragraphs.                                                                                   |  |  |  |
| Line 31, framboids are present in sulphides and oxides.                                                                                                                                                                                                                                                                                                                                       | The sentence has been improved to be clearer.                                                 |  |  |  |
| Line34, spelling on euxinic                                                                                                                                                                                                                                                                                                                                                                   | The spelling has been corrected.                                                              |  |  |  |
| Line 35, the text here is not accurate. Pyrite
framboids are markers of the redox transition
between oxygen-containing and
anaerobic/sulphidic waters. In fact, some pulse of
an oxidant is needed for pyrite formation since its
sulfur atoms are present in the -1-valence state
and precursor minerals like FeS are in the -2-
valence state.                         | Thank you for clarifying this issue. We included this in the final version of the manuscript. |  |  |  |
| It is true that the Ohfuji paper looked at abiotic
synthesis, but so did many other studies (Farrand
et al. 1970; Berner 1969 Econ Geol v. 64;
Graham and Ohmoto 1994 Geochim Cosmochim
v. 58; Sweeney and Kaplan 1973 Econ Geol;
Wilkin and Barnes 1996 Geochim. Cosmochim
v. 60). These should be referenced, at a minimum,
along with others. Seems to me that a very | Thank you for providing additional references.
We have included them in the manuscript.    |  |  |  |

**Reviewer** 1

| accurate and trustworthy evaluation of abiotic
syntheses is required here as a point of
comparison.                                                                                                                                                                                                                                                                                                                                          |                                                                                                                                                                                                                                                                                                                                                                                                                                                                                                                                                                                                                                                                                                                                                                                                                            |
|----------------------------------------------------------------------------------------------------------------------------------------------------------------------------------------------------------------------------------------------------------------------------------------------------------------------------------------------------------------------------------------------------------------------------------------------------|----------------------------------------------------------------------------------------------------------------------------------------------------------------------------------------------------------------------------------------------------------------------------------------------------------------------------------------------------------------------------------------------------------------------------------------------------------------------------------------------------------------------------------------------------------------------------------------------------------------------------------------------------------------------------------------------------------------------------------------------------------------------------------------------------------------------------|
| Line 42, why Interestingly?                                                                                                                                                                                                                                                                                                                                                                                                                        | We think there was a misuse of the word. The sentence was rewritten.                                                                                                                                                                                                                                                                                                                                                                                                                                                                                                                                                                                                                                                                                                                                                       |
| Line 53, why should it be assumed? The Introduction is very fuzzy about the sizes and dimensions of bacterial viruses. The 50 to 200 nm size range makes to connection to any dimension in framboids that is consistent across the environments of their formation. And the image shown in Figure 2 is not connective to framboid sizes or shapes. The whole set up seems to be a stretch.                                                         | We have rearranged the paragraph and added an
additional paragraph (1.1) that includes more
information about viruses and their involvement
in ecosystems, as well as the behaviour.
Please note that Figure 2 shows the stained
bacteriophages and not the structures obtained
during this study.                                                                                                                                                                                                                                                                                                                                                                                                                                                                                                       |
| Framboids are quite notably present in
hydrothermal deposits yet this fact is quite
conveniently omitted. Hydrothermal occurrence
should be noted as a fact that may be contrary to
bacteriophage involvement.                                                                                                                                                                                                                         | We have included additional references.
We can agree that hydrothermal occurrence of
framboidal pyrite may stand against the
bacteriophage involvement, but bacteriophages
(or Archeoviruses) may also be present in such
environments (up to 120°C). We have also
included additional references related to this
topic. However, the issue is still being
investigated.                                                                                                                                                                                                                                                                                                                                                                                                                           |
| The experimental setup lacks any connection to
natural settings and does not recognize the
established understanding of pyrite formation. 1)
describe redox and pH control; 2) describe aging.
The use of the culture medium is an experimental
requirement, perhaps. But it should be
acknowledged that you are setting up the
experiment to favor biology and not reproduce
anything that happens in nature.             | We are aware that the experimental setup is far
from natural conditions. However, since the
study is a preliminary project, we focuses on the
setup where most conditions can be measured or
standardized. An extension of the study is
required, e.g., ionic strength or Eh were not
considered in this approach (but were controlled
during the experiment).                                                                                                                                                                                                                                                                                                                                                                                                                                        |
| I was left unsure why so much emphasis was
placed on the zeta potential experiments & data.
Provide some introduction as to why this is
important.                                                                                                                                                                                                                                                                                        | We included information about $\zeta$ -potential and
the importance of the measurements (see 1.1.
section)                                                                                                                                                                                                                                                                                                                                                                                                                                                                                                                                                                                                                                                                                                           |
| Results: the XRD data show that the experiments
produced nothing that is realistic for natural
settings. The traces of pyrite in the XRD patterns
are not especially characteristic or convincing.
Troilite is a high temperature phase; there should
be no expectation for troilite identification. How
were the samples collected for XRD studies?
Please describe the controls that were in place to
prevent oxidation. | We have added some explanations to the methodology in this regard. To avoid the oxidation of the samples, a glovebox with $N_2$ atmosphere was used. Moreover, the investigated samples were packed into the capillaries. The lack of characteristic pick of pyrite in XRD patters may be induced by sample oxidation and masked by substrates used for the synthesis experiments. Yes, troilite is a high temperature phase, and we do not have any information how the viruses may indicate the crystallization of this phase due to this is a preliminary study or it is miss identification of this phase. We know that troilite is unlikely to form in the experimental setup. However, in quite noisy diffraction patterns, the identification of individual phases may be erroneous. We decided that if the adopted |

|                                                                                                                                                                                                                                                                                                                                                                                                                                                                                        | identification algorithm shows such a phase, we
place it on the diffractogram. Due to our fault,
there was no additional comment that should be
included in the manuscript, as suggested by the
reviewer. In addition, research on this type of
material is very troublesome. Sulfide minerals
precipitated in the presence of biological material
such as bacteria or phages seem to be extremely
susceptible to oxidation. Despite the use of a
nitrogen-filled chamber and XRD capillaries, we
did not avoid the formation of oxidation products
(see discussion, lines 282 - 290).                                                                                                                                                                                                                                                                                 |
|----------------------------------------------------------------------------------------------------------------------------------------------------------------------------------------------------------------------------------------------------------------------------------------------------------------------------------------------------------------------------------------------------------------------------------------------------------------------------------------|---------------------------------------------------------------------------------------------------------------------------------------------------------------------------------------------------------------------------------------------------------------------------------------------------------------------------------------------------------------------------------------------------------------------------------------------------------------------------------------------------------------------------------------------------------------------------------------------------------------------------------------------------------------------------------------------------------------------------------------------------------------------------------------------------------------------------------------------------------------------------------------------------------|
| In the SEM studies, did you use backscatter detection? This would have revealed the high atomic Z particles more precisely.                                                                                                                                                                                                                                                                                                                                                            | During this study we used the InLens and BSE detector.                                                                                                                                                                                                                                                                                                                                                                                                                                                                                                                                                                                                                                                                                                                                                                                                                                                  |
| The images shown in Figure 6 look less like
framboids then the kinds of precipitates that form
in abiotic experiments. There is no improvement
in morphological connection to framboid
structures, in fact a step back, which leaves the
reader unconvinced that this represents anything
new. The weak similarity should be
acknowledged.                                                                                                                        | Our research is certainly inconclusive and
uncertain whether phages can influence the
formation of framboids. However, we clearly
show that pages influence e.g., the size of the
mineral phases formed in their presence. This
effect is very clearly visible with the naked eye
during experiments, which we pay attention to
especially during the precipitation of copper
sulphides. In addition, phages affect the degree of
agglomeration of fine crystallites, possibly
favouring the formation of spherical forms. We
do not claim, however, that the structures we
obtained are framboids, hence we always use the
phrase "framboid-like". We give rise to further
discussion on this topic. This is only the first such
experimental study to discuss the role of phages
in the formation of sulphide minerals at all (see
lines 334-340). |
| Line 270, microcrystals in framboids come in
multiple shapes and sizes. The link to Ohfuji and
Akai is a weak thread. Please acknowledge the
full range of microcrystal sizes and shapes.
Line 277, pyrite framboid size in nature depends
on the environment of formation (it is not based
on the study - not sure what this means). A
coherent model would need to explain why
viruses would produce different sized framboids
in different environments. | We have rearranged the paragraph to make it
clearer. We have also included an extra table
(Table 1, page 10) with more information about
sizes (please find Table 1 attached at the end of
the document). The information about the shape
of microcrystals has been also included in the
manuscript (see lines 308-311).                                                                                                                                                                                                                                                                                                                                                                                                                                                                                                                                                              |
| Line 284, again there are many abiotic framboid
synthesis examples, all of them produce even
more framboid-like morphology than what is
described here (see references above in point 5;
please consider adding these to the discussion for
an objective analysis).                                                                                                                                                                                                     | We have rearranged the whole paragraph and
included a table with an overview of
experimental conditions (Table 2, page 11)
(please find Table 2 attached at the end of the
document).
We have also included the publications
mentioned in Point 5.                                                                                                                                                                                                                                                                                                                                                                                                                                                                                                                                                                                                                                    |

**Reviewer 2**

| Comments                                                                                                                                                                                                                                                                                                                                                                                                                                                                     | Anwsers                                                                                                                                                                                                                                                                                                                                                                                                                                                                                                                                                                                                                                                                                                                                                                                                                                                                                                                                                                                                                                                                     |
|------------------------------------------------------------------------------------------------------------------------------------------------------------------------------------------------------------------------------------------------------------------------------------------------------------------------------------------------------------------------------------------------------------------------------------------------------------------------------|-----------------------------------------------------------------------------------------------------------------------------------------------------------------------------------------------------------------------------------------------------------------------------------------------------------------------------------------------------------------------------------------------------------------------------------------------------------------------------------------------------------------------------------------------------------------------------------------------------------------------------------------------------------------------------------------------------------------------------------------------------------------------------------------------------------------------------------------------------------------------------------------------------------------------------------------------------------------------------------------------------------------------------------------------------------------------------|
| Lines 14-16: Please re-write or delete this phrase.
In the context of biomineralization, ions are not
attracted by biological surfaces – or are capsids
electrically charged, or magnetic? Please explain
how could a virus change the electrochemical
properties of precipitated minerals.                                                                                                                                                                   | We have deleted the misleading part of the phrase. We think that the net surface charge of the minerals precipitated with viruses can be changed. We included it in the discussed section (see lines 275-278).                                                                                                                                                                                                                                                                                                                                                                                                                                                                                                                                                                                                                                                                                                                                                                                                                                                              |
| Line 18: They are different viruses, and thus it was
expected different outcomes in the experiments.
The presence of a lipid envelope is certainly a
difference between the two viruses, but note that
cells have membranes and are frequently involved
in biomineralization.
Lines 21-22: (i) see comments for lines 14-16; (iii)
what colloid? The structure of framboids indicate
they form as a sphere of crystallites, not by
agglomeration. | We have rearranged the sentence. We are aware
cells have lipid membranes that can be involved
in biomineralization. We just wanted to point
out that the lack of a lipid membrane in the case
of phages might have an impact on the
formation of framboid-like structures, but we
do not claim that this factor is crucial.
We assumed that viruses could facilitate the
formation of tiny crystallites, that could form
more complex forms at a later stage. We do not
state that framboids arise from agglomeration.
We argue that viruses, e.g., by changing the net
surface charge, can favour the formation of very
fine crystallites. This process can be seen with
the naked eye - the presence of viruses causes
the formation of very fine structures (this fact it
is even used in the synthesis of nanostructures).
In contrast, virus-free synthesis produces a
massive precipitate. We have changed the
description of the process and we additionally
discussed this subject (see lines 334-340). |
| Lines 28-29: Microbes can oxidize both sulfide and Fe(II) from pyrite and other sulfide minerals, producing sulfate, Fe(III) minerals and protons, leading to decrease in pH.                                                                                                                                                                                                                                                                                                | We have supplemented the sentence with additional information (see lines 49-50).                                                                                                                                                                                                                                                                                                                                                                                                                                                                                                                                                                                                                                                                                                                                                                                                                                                                                                                                                                                            |
| Lines 60-65: Please re-write these sentences. The
hypothesis/objectives are not clear. It is unclear
what kind of experiments were done.
In addition, the importance of your findings is
over-emphasized – data presented are not sufficient
to access the importance of the formation of
framboids associated to viruses in nature.                                                                                                                       | We have rewritten the paragraph.
Please note, that we do not state that the
structures obtained are framboids, but
framboid-like. We also suggest that
bacteriophages might be a factor in one of the
ways of framboid-like structures formation. We
additionally discussed that problems (see lines
334-340.                                                                                                                                                                                                                                                                                                                                                                                                                                                                                                                                                                                                                                                                                                                                          |
| Lines 111-117: Were the solutions treated to
remove dissolved oxygen? How? Was O2 in the
solutions quantified?                                                                                                                                                                                                                                                                                                                                                         | We have included the information. It was
omitted by our mistake.
The concentration of oxygen in the glovebox
was controlled with an oxygen meter. The
water prior to use was degassed by
autoclaving (121°C, 15 min).                                                                                                                                                                                                                                                                                                                                                                                                                                                                                                                                                                                                                                                                                                                                                                                                                                        |
| Lines 127-128: ions are not attracted by viruses:
they adsorb, bind, etc.
Line 130: Were the viral capsids isolated? How?
How binding of Fe, Cu and sulfide to viruses were
measured?                                                                                                                                                                                                                                                                            | We changed the misleading word to "adsorb" in
the whole manuscript.
The viral capsids were not isolated. The whole
solution was placed in the measurement cell.
The apparatus was used to measure the $\zeta$ -
potential of viruses with adsorbed cations or                                                                                                                                                                                                                                                                                                                                                                                                                                                                                                                                                                                                                                                                                                                                                                                                |

|                                                                                                                                                                                                                                                      | anions (change in charge of electrical double layers).                                                                                                                                                                                          |
|------------------------------------------------------------------------------------------------------------------------------------------------------------------------------------------------------------------------------------------------------|-------------------------------------------------------------------------------------------------------------------------------------------------------------------------------------------------------------------------------------------------|
|                                                                                                                                                                                                                                                      | A sentence describing the process was added.
"Subsequently, the solutions were transferred to
measurement cells and measured."                                                                                                            |
| Lines 139-143: Please explain how samples were
prepared for XRD. Of crucial importance is how
samples were stored, and how long they were
stored.                                                                                           | We have included additional information. We
put great effort to prevent the oxidation of the
samples. However, the transfer of the samples
to the capillaries caused partial oxidation. We
pointed out this problem in the results. |
| Line 141: there is no "CoK $\alpha$ lamp" in X-ray                                                                                                                                                                                                   | We have changed "CoK $\alpha$ lamp" to "CoK $\alpha$                                                                                                                                                                                            |
| diffractometers.                                                                                                                                                                                                                                     | radiation source".                                                                                                                                                                                                                              |
| coverslips are widely used to mount samples for SEM.                                                                                                                                                                                                 | we use glass slides due to their better
mechanical properties. Our protocol was
adjusted for glass slides. Glass coverslips are
very fragile and may break during the
procedure.                                                    |
| Line 148: Are you sure it was 20 nm? Or was this an estimate from the device?                                                                                                                                                                        | The layer was 20 nm thick, and it was applied by a device.                                                                                                                                                                                      |
| Lines 170-171: are differences in the samples                                                                                                                                                                                                        |                                                                                                                                                                                                                                                 |
| with and without viruses (or bacteriophages). Were
the differences noted in the z-potential or the
conductivity?
Lines 171-172: were the significant differences
found between the FeS and CuS groups, or among                          | We changed the sentence to be clearer. The significance test was performed among the same group of metals.                                                                                                                                      |
| the three treatments for the same metal?                                                                                                                                                                                                             |                                                                                                                                                                                                                                                 |
| Lines 177-178: Was the z-potential in the last column measured with virus + metal + Na2S, or with virus + Na2S?                                                                                                                                      | We have clarified the sentence. The Na 2 S (similarly to FeSO 4 or CuSO 4 ) was added as sole solution (see lines 195-198)                                                                                     |
| Lines 188-194: Any differences in the mineral composition were probably masked by sample oxidation. This is stated in lines 193-194. But the sentence in line 188 states that "Phase composition of samples did not differ significantly". You could | Mohrite, Butlerite and jarosite are the product
of oxidation. The list of the phases which are
results of oxidation of the sample was added to
the text.                                                                               |
| re-arrange the text to make clear in the beginning
of the paragraph that the X-ray data should be
analyzed knowing that the sample has been
oxidized.                                                                                       | Substrates: mohrite
Oxidation product: butlerite, chalkantite,
kröhnkite, natrochalcite, jarosite                                                                                                                                         |
| Are mohrite, blutlerite and jarosite products of oxidation of the sample? You should make a list of                                                                                                                                                  | Sulphides: troilite, pyrite                                                                                                                                                                                                                     |
| probable original products and oxidation products
for the FeS and CuS experiment, to help the reader
unfamiliar with these minerals.                                                                                                           | We added the explanation in 3.3. section and we discussed this issue (see lines 282-290)                                                                                                                                                        |
| Lines 197-198: How did you recognize the mineral                                                                                                                                                                                                     |                                                                                                                                                                                                                                                 |
| phases in the SEM to record the EDS spectra?                                                                                                                                                                                                         |                                                                                                                                                                                                                                                 |
| Maybe the use of the term "mineral phase" is a                                                                                                                                                                                                       | There was a misuse of the term "mineral                                                                                                                                                                                                         |
| narticles" or something like that?                                                                                                                                                                                                                   | pnases . we have changed the term to minerals                                                                                                                                                                                                   |
| Line 200: the use of "mineral phases" is probably                                                                                                                                                                                                    | in the entire manuscript.                                                                                                                                                                                                                       |
| a mistake here too.                                                                                                                                                                                                                                  |                                                                                                                                                                                                                                                 |

| Lines 200-201: The spectra "1" and "2" seem very similar, not distinct.                                                                                                                                                                                                                                                                                                                                                                                                                                                                                                                                                                                                                                                    | We changed the sentence to: "Spectra 1 and 2
revealed clearly visible signals from iron and
sulphur."                                                                                                                                                                                                                                                                                                                                                                                                                                                                                                                                                                                                                                                                       |
|----------------------------------------------------------------------------------------------------------------------------------------------------------------------------------------------------------------------------------------------------------------------------------------------------------------------------------------------------------------------------------------------------------------------------------------------------------------------------------------------------------------------------------------------------------------------------------------------------------------------------------------------------------------------------------------------------------------------------|-----------------------------------------------------------------------------------------------------------------------------------------------------------------------------------------------------------------------------------------------------------------------------------------------------------------------------------------------------------------------------------------------------------------------------------------------------------------------------------------------------------------------------------------------------------------------------------------------------------------------------------------------------------------------------------------------------------------------------------------------------------------------------------|
| Line 204: O and Na are not negligible in spectrum 4, and Na is not negligible in spectrum 5.                                                                                                                                                                                                                                                                                                                                                                                                                                                                                                                                                                                                                               | The presence of O, and Na in spectrum 4, and
Na in spectrum 5 is not negligible. However, the
analytical line of Na K $\alpha$ is overlapping with the
Cu L 1 line so precise estimation of how much
Na is in the sample is very hard even if the
deconvolution of the analytical lines is used.
Moreover, it is very hard to tell if the presence
of Na in the investigated crystallites due to the
electron beam may also excite phases below the
cooper sulfate crystallites                                                                                                                                                                                                                                                             |
| Lines 209-216: The text is very confusing. From
the images in Figure 6, it seems that the shapes and
sizes of the particles are not really different in the
three first groups of images (a-l). In figures m-p
they seem smaller, and sometimes are arranged in
spheres.
Line 210: "Experiments with bacteriophages gave
similar structures". Similar to what? To each other?
Lines 222-223: From the images, images in Figure
7 not seem to be really different from images in
Figure 6 without bacteriophages.
Lines 234-235: The differences observed with the
naked eye must be diverse from that observed at
the SEM – the magnification are orders of
magnitude different. | We would like to apologize if we were not very
clear. We wanted to show that phages cause tiny
crystallites to aggregate into spherical
structures frequently. In addition, in
experiments with phages, many fine 200-300
nm crystallites are formed, which are absent in
abiotic experiments. We indicate that the
differences are not significant, but quite
pronounced, especially with the P1 phage and
with mixing vs. no mixing. Spherical structures
never arise in mixing experiments, not even
with phages. In the case of the experiment with
Phi6 phage (Fig. 7) we note in the text that there
are practically no visible differences here. Only
an overrepresentation of very fine minerals can
be seen, similar to P1. |
| Line 242: Please provide a reference for the DLVO theory.                                                                                                                                                                                                                                                                                                                                                                                                                                                                                                                                                                                                                                                                  | We have also included the reference regarding DLVO theory.                                                                                                                                                                                                                                                                                                                                                                                                                                                                                                                                                                                                                                                                                                                        |
| Lines 242-243: Again: viruses are not nuclei which
attract ions from solution. If they attracted ions
from solution, they would be soon encased in
minerals and would not be effective in infecting
cells (which are the sine-qua-non condition for
virus replication).                                                                                                                                                                                                                                                                                                                                                                                                                                     | We have changed the misused word to "bind".
Kindly note, that we are aware that viruses can
infect bacterial cells only under proper
conditions, when capsids are not mineralized.
But what is important, even when the infection
takes place, bivalent ions (like Mg 2+ ) are
needed. However, in this research we did not
study the process of infection when the
saturation index is exceeded.                                                                                                                                                                                                                                                                                                                                              |
|  <li>Line 244: the work cited states that capsids can bind iron and nucleate iron minerals.</li> <li>Line 249 Viruses do not attract ions; they can bind, or adsorb, ions.</li> <li>Lines 251-252: bind, not attract. If they bind, you can measure differences relatively to the sample without Na2S.</li>                                                                                                                                                                                                                                                                                                                                                                                                       | As we mentioned, the misleading word "adsorb" has been changed in the whole manuscript.                                                                                                                                                                                                                                                                                                                                                                                                                                                                                                                                                                                                                                                                                           |
| Lines 254-255: It is probably much more complex
than just charge interactions. For example, sulfide
ions may interact with -SH groups of cysteine
residues.                                                                                                                                                                                                                                                                                                                                                                                                                                                                                                                                                       | We are aware that there can be many other
phenomena. We have included a hedged
statement.                                                                                                                                                                                                                                                                                                                                                                                                                                                                                                                                                                                                                                                                                   |
| Line 261: Are the two groups statistically different
from each other? Or each experiment is different
from the others in the same group?                                                                                                                                                                                                                                                                                                                                                                                                                                                                                                                                                                             | The statistical significance test was performed
among the same metal group. We have
rewritten the sentence to remove the ambiguity.                                                                                                                                                                                                                                                                                                                                                                                                                                                                                                                                                                                                                                         |

| Lines 268-268: The measurements are not
erroneous if the effects of aggregation inherent to
the technique are considered in interpretation. Even
if the numerical results do not represent real sizes,
they can still be used to compare samples.                                                                                                                                                                                                                                                                                                                   | Yes, we agree with your comment. However,
we wanted to note, that the results may be
somehow erroneous.                                                                                                                                                                                                                                                                                                                                                                                                                                                                                          |
|---------------------------------------------------------------------------------------------------------------------------------------------------------------------------------------------------------------------------------------------------------------------------------------------------------------------------------------------------------------------------------------------------------------------------------------------------------------------------------------------------------------------------------------------------------------------------------|--------------------------------------------------------------------------------------------------------------------------------------------------------------------------------------------------------------------------------------------------------------------------------------------------------------------------------------------------------------------------------------------------------------------------------------------------------------------------------------------------------------------------------------------------------------------------------------------------------|
| 271-273: Was the P1 virus chosen because of the icosahedral shape, or the icosahedral shape was a coincidence? Several minerals can show icosahedral morphologies, including macroscopic specimens of natural pyrites. Crystal shapes depend mainly on the arrangement of atoms, and this is the case of icosahedral pyrites. Conversely, there are several icosahedral viruses. The fact that pyrite in framboids and in your experiments is icosahedral viruses, since they can be produced by purely chemical processes. Lines 300-302: The idea or virus capsids attracting | We do not state that the icosahedral shape of the virus is a sine qua non for framboidal pyrite formation. The P1 virus was chosen as an example of a common family of "crystalline" shaped viruses. The Phi6 virus was selected as an example of a lipid enveloped virus. We did not know in advance what effect such a difference could have. Most likely, the two viruses have different effects on mineral precipitation, possibly due to the presence or absence of a lipid envelope. We indicate this effect, but it requires more careful study. We have changed the misused word in the |
| ions again – please consider that binding is the
important thing to consider for mineral
precipitation. To precipitate a mineral, it is needed                                                                                                                                                                                                                                                                                                                                                                                                                            | whole manuscript. We have also considered all your comments on this issue.                                                                                                                                                                                                                                                                                                                                                                                                                                                                                                                             |
| several layers of each type of ion.
Lines 318-319: Again the idea of viruses attracting
ions from solution. Here, attract and bind seem to
have been used as synonyms. Are you sure that the
lipid envelope is the cause of the differences
observed between the viruses? They surely have
other differences too.                                                                                                                                                                                                                                             | We are not sure about the influence of the lipid
envelope. Therefore, we only assumed that the
lipid envelope is a differentiating factor. We
aimed to check this phenomenon, and thus we
chose two morphologically different
bacteriophages in the preliminary studies.                                                                                                                                                                                                                                                                                                                |
| Lines 322-323: Fe(II) and S(II-) are not oxidized in
anoxic environments, and FeS precipitation is not
limited in these environments. Most of the Earth's
crust is anoxic, including deep soils and sediments.                                                                                                                                                                                                                                                                                                                                                         | Thank you for your comment. We are aware of
this problem. The ambiguous sentence was
changed in the final version of the manuscript.                                                                                                                                                                                                                                                                                                                                                                                                                                                             |
| Figure 2: Please provide scale bars for the light
micrographs.
Figure 2 (caption): Magnification in printed
micrographs is almost useless – it depends on how
much the image is enlarged. Please write the
bacterial genus Pseudomonas in italics. Please
state in the caption the technique used
(fluorescence light microscopy or transmission
electron microscopy) in each image.                                                                                                                                                                    | The scale bars have been added. We have changed the captions.                                                                                                                                                                                                                                                                                                                                                                                                                                                                                                                                          |
| Figure 3b: Please make a legend without colors for
Figure 3b, to make it clear the differences between
conductivity and z-potential in the graph.                                                                                                                                                                                                                                                                                                                                                                                                                         | This was changed in the final version of the manuscript.                                                                                                                                                                                                                                                                                                                                                                                                                                                                                                                                               |
| Figure 3 (caption): Please explain what where the solutions used for measurements (phosphate buffer, saline, etc) in each plot.
In (b), there is a plot of conductivity which is not mentioned in the caption                                                                                                                                                                                                                                                                                                                                                                | We have included additional information
regarding the solution used, as well as
information about conductivity. All the arisen
issues were changed in the final version of the
manuscript.                                                                                                                                                                                                                                                                                                                                                                                                 |
| In (c), it is shown the z-potential of complex materials in suspension, not the "attraction of ions                                                                                                                                                                                                                                                                                                                                                                                                                                                                             | (**) means statistical differences among datasets represented on bars in Figure b and c (mean value from 3 measurements).                                                                                                                                                                                                                                                                                                                                                                                                                                                                              |

| by bacteriophages" – there are other ingredients in the mixture.                                                                                                                                                                                                                                                                                                                                                                                                                                                                                                                                                                                                                                                                                                                        |                                                                                                                                                                                                                                                                                                                                                                                                                                                                                                                                                                                                                                                                                                                                                                                                                                                                                                                                                                                                                                           |
|-----------------------------------------------------------------------------------------------------------------------------------------------------------------------------------------------------------------------------------------------------------------------------------------------------------------------------------------------------------------------------------------------------------------------------------------------------------------------------------------------------------------------------------------------------------------------------------------------------------------------------------------------------------------------------------------------------------------------------------------------------------------------------------------|-------------------------------------------------------------------------------------------------------------------------------------------------------------------------------------------------------------------------------------------------------------------------------------------------------------------------------------------------------------------------------------------------------------------------------------------------------------------------------------------------------------------------------------------------------------------------------------------------------------------------------------------------------------------------------------------------------------------------------------------------------------------------------------------------------------------------------------------------------------------------------------------------------------------------------------------------------------------------------------------------------------------------------------------|
| In (d), it is shown the size of FeS and CuS particles.
Does ** means statistical differences? It is not                                                                                                                                                                                                                                                                                                                                                                                                                                                                                                                                                                                                                                                                              |                                                                                                                                                                                                                                                                                                                                                                                                                                                                                                                                                                                                                                                                                                                                                                                                                                                                                                                                                                                                                                           |
| clear what data were compared in statistical tests.                                                                                                                                                                                                                                                                                                                                                                                                                                                                                                                                                                                                                                                                                                                                     |                                                                                                                                                                                                                                                                                                                                                                                                                                                                                                                                                                                                                                                                                                                                                                                                                                                                                                                                                                                                                                           |
| Figure 4: Please highlight the peaks for the                                                                                                                                                                                                                                                                                                                                                                                                                                                                                                                                                                                                                                                                                                                                            |                                                                                                                                                                                                                                                                                                                                                                                                                                                                                                                                                                                                                                                                                                                                                                                                                                                                                                                                                                                                                                           |
| preparation.                                                                                                                                                                                                                                                                                                                                                                                                                                                                                                                                                                                                                                                                                                                                                                            |                                                                                                                                                                                                                                                                                                                                                                                                                                                                                                                                                                                                                                                                                                                                                                                                                                                                                                                                                                                                                                           |
| Figure 4 (caption): Please separate Fe and Cu
minerals in the list at the end of the caption, and
provide a chemical formula for each mineral. You
can see mindat.org for mineral formulae. In the
figure, is "Ja" for jarosite?
Figure 5: The lettering in the scale bars are too
small. Please increase them.                                                                                                                                                                                                                                                                                                                                                                                                                                                       | Thank you for your comment. It was corrected
in the final version of the manuscript.
"Ja" is the description for jarosite. This was
mistakenly omitted in the caption.                                                                                                                                                                                                                                                                                                                                                                                                                                                                                                                                                                                                                                                                                                                                                                                                                                                           |
| Figure 5 (caption): How was this material                                                                                                                                                                                                                                                                                                                                                                                                                                                                                                                                                                                                                                                                                                                                               | These are only sample images, and the EDS                                                                                                                                                                                                                                                                                                                                                                                                                                                                                                                                                                                                                                                                                                                                                                                                                                                                                                                                                                                                 |
| prepared? With viruses, or not?                                                                                                                                                                                                                                                                                                                                                                                                                                                                                                                                                                                                                                                                                                                                                         | spectra were used to show how we tentatively
differentiated the mineral structures under
SEM. As is well known, mineral structures are
not always easy to distinguish under SEM, but
using EDS spectra it is easy to determine
whether we are dealing with sulphides or with
sulphates and other oxidation products. This is
what EDS spectra served, and that is why we
show them only as an example.                                                                                                                                                                                                                                                                                                                                                                                                                                                                                                                                                                                                            |
| Figures 6 and 8 (captions): It is mentioned in the                                                                                                                                                                                                                                                                                                                                                                                                                                                                                                                                                                                                                                                                                                                                      | We are sorry. That was our mistake. This is an                                                                                                                                                                                                                                                                                                                                                                                                                                                                                                                                                                                                                                                                                                                                                                                                                                                                                                                                                                                            |
| captions that the experiments were made with the                                                                                                                                                                                                                                                                                                                                                                                                                                                                                                                                                                                                                                                                                                                                        | unclear caption under the figure. Of course, the                                                                                                                                                                                                                                                                                                                                                                                                                                                                                                                                                                                                                                                                                                                                                                                                                                                                                                                                                                                          |
| P1 bacteriophage, but a-d and i-l were prepared                                                                                                                                                                                                                                                                                                                                                                                                                                                                                                                                                                                                                                                                                                                                         | experiment was performed with phages and                                                                                                                                                                                                                                                                                                                                                                                                                                                                                                                                                                                                                                                                                                                                                                                                                                                                                                                                                                                                  |
| without bacteriophages.                                                                                                                                                                                                                                                                                                                                                                                                                                                                                                                                                                                                                                                                                                                                                                 | without phages as a control.                                                                                                                                                                                                                                                                                                                                                                                                                                                                                                                                                                                                                                                                                                                                                                                                                                                                                                                                                                                                              |
| Figure 10: The idea of icosahedral viruses serving
as nuclei for the synthesis of icosahedral FeS
mineral particles, expressed in the drawings and
also in the text, is not based on your data, nor on
data from the literature. It seems pure imagination.
A second issue is that there are two possibilities for
formation of spherical structures: agglomeration as
suggested in Figure 10, or they are formed already
as a sphere. For framboidal pyrite, it seems more
likely that it forms as a sphere: otherwise, how
could they be so well organized by aggregation of
pre-existing particles? The "framboids" shown in
other figures of this manuscript are too smooth to
have been formed by aggregation of pre-existing
particles. | We did not want to confuse the readers. If we
were not clear enough, we would like to
apologize. We do not claim anywhere directly
that the icosahedral structure of the virus causes
the formation of framboid pyrite. We are merely
claiming that the "crystalline" structure of
viruses such as P1, which happens to be
icosahedral, but does not to be so, is likely to
facilitate the formation of some minerals or
their crystalline forms.
The angular structure of the virus in this figure
corresponds to the above-mentioned problem
but does not indicate the icosahedral structure.
The problem described here has been recently
discussed e.g., in these papers:
Słowakiewicz, M., Borkowski, A., Syczewski,
M. D., Perrota, I. D., Owczarek, F., Sikora, A.,
Detman, A., Perri, E., and Tucker, M. E.:
Newly-discovered interactions between
bacteriophages and the process of calcium
carbonate precipitation, Geochim. Cosmochim.
Acta, 292, 482–498, 2021 |
|                                                                                                                                                                                                                                                                                                                                                                                                                                                                                                                                                                                                                                                                                                                                                                                         | Perri, E., Tucker, M. E., Słowakiewicz, M.,
Whitaker, F., Bowen, L., and Perrotta, I. D.:
Carbonate and silicate biomineralization in a                                                                                                                                                                                                                                                                                                                                                                                                                                                                                                                                                                                                                                                                                                                                                                                                                                                                                             |

|                                                                                                                                                                                                                                                                     | hypersaline microbial mat (Mesaieed sabkha,
Qatar): Roles of bacteria, extracellular
polymeric substances and viruses,
Sedimentology, 65, 1213–1245, 2018.
Perri, E., Słowakiewicz, M., Perrotta, I. D., and
Tucker, M. E.: Biomineralization processes in
modern calcareous tufa: Possible roles of
viruses, vesicles and extracellular polymeric
substances (Corvino Valley – Southern Italy),
Sedimentology, 69, 399–422, 2022. |  |  |
|---------------------------------------------------------------------------------------------------------------------------------------------------------------------------------------------------------------------------------------------------------------------|---------------------------------------------------------------------------------------------------------------------------------------------------------------------------------------------------------------------------------------------------------------------------------------------------------------------------------------------------------------------------------------------------------------------------------------------------------------|--|--|
| Technical comments:                                                                                                                                                                                                                                                 |                                                                                                                                                                                                                                                                                                                                                                                                                                                               |  |  |
| Lines 13 and 19: do you mean Enterobacter, or
enterobacteria? If it is Enterobacter (genus), it
should be written in italics; if it is enterobacteria
(common word), it should be written without
capital letter.                                       | According to DSMZ, we changed the name to
"Escherichia phage P1" in the whole manuscript.
https://www.dsmz.de/collection/catalogue/
details/culture/DSM-5757                                                                                                                                                                                                                                                                                  |  |  |
| Lines 14 and 19: Pseudomonas should be written
in italics, since it is the name of a bacterial genus                                                                                                                                                             | The name has been marked in italics.                                                                                                                                                                                                                                                                                                                                                                                                                          |  |  |
| Line 29: sulphide.
Line 33: Please add a space before the parenthesis.
Is it greigite?
Line 34: euxinic?                                                                                                                                                   | The spellings have been corrected.                                                                                                                                                                                                                                                                                                                                                                                                                            |  |  |
| Line 40: I suggest to begin a new paragraph to
explain the basics of viruses and bacteriophages.
Lines 42-43: the same idea is expressed better in
lines 51-53.
Line 49: I suggest to begin a new paragraph to
present the phages used in this work. | The paragraph has been rewritten. Additional information about viruses has been added. We have included information about their life modes, types of infections, and the impact on the environment.                                                                                                                                                                                                                                                    |  |  |
| Lines 51-53: Consider using "can" only once.                                                                                                                                                                                                                        | The sentence has been rewritten.                                                                                                                                                                                                                                                                                                                                                                                                                              |  |  |
| Lines 60-61: I think "in this work" would be better.                                                                                                                                                                                                                | "In this paper" has been replaced with "in this work"                                                                                                                                                                                                                                                                                                                                                                                                         |  |  |
| Line 75: laminar flow cabinet?                                                                                                                                                                                                                                      | "Laminar chamber" has been replaced with "laminar flow cabinet"                                                                                                                                                                                                                                                                                                                                                                                               |  |  |
| Line 89: You may use "used" instead of "added".                                                                                                                                                                                                                     | "Added" has been replaced with "used"                                                                                                                                                                                                                                                                                                                                                                                                                         |  |  |
| Line 137: the device.                                                                                                                                                                                                                                               | The preposition has been included.                                                                                                                                                                                                                                                                                                                                                                                                                            |  |  |
| Line 164: "both" is not suitable here.                                                                                                                                                                                                                              | We have changed "both" for "studied".                                                                                                                                                                                                                                                                                                                                                                                                                         |  |  |
| Line 192: There is a misspelling here, please write "chalcanthite".                                                                                                                                                                                                 | The misspelling has been corrected.                                                                                                                                                                                                                                                                                                                                                                                                                           |  |  |
| Figure 4, caption: There is some misspelling:
synthesized, in the mineral names, please write
"chalcanthite" and "troilite" (you can see
mindat.org for correct mineral names).                                                                            | :
We have checked the names on the provided
website. The misspellings have been corrected                                                                                                                                                                                                                                                                                                                                                               |  |  |
| Lines 197-198: EDS spectra are not measured, they are obtained.                                                                                                                                                                                                     | We have changed "measured" to "obtained".                                                                                                                                                                                                                                                                                                                                                                                                                     |  |  |
| Lines 209-210: visibly small
Line 244: viral                                                                                                                                                                                                                     | The spelling has been corrected.                                                                                                                                                                                                                                                                                                                                                                                                                              |  |  |
| Lines 287-288: the hydration of the FeSO4 is excessive detail.                                                                                                                                                                                                      | We have removed excessive details.                                                                                                                                                                                                                                                                                                                                                                                                                            |  |  |
| Lines 295-297: Try writing a single phrase
comparing your results with others'.
Lines 298-299: Try rewriting this phrase using
"stir" only once.                                                                                                           | $\frac{1}{2}$ We have rewritten the sentences.                                                                                                                                                                                                                                                                                                                                                                                                                |  |  |

Table 1.

| Microcrystal
diameter
[µm] | Framboid
diameter
[µm] | Source                                                                    | Reference                                               |
|----------------------------------|------------------------------|---------------------------------------------------------------------------|---------------------------------------------------------|
| 2-3                              | -                            | Precambrian rocks with copper and lead-zinc
ore; Mount Isa Shale       | (Love and Zimmerman,
1961)                           |
| 0.12                             | 12                           |                                                                           |                                                         |
| 0.9                              | 10                           | Deen ees estimentes Annale Deein                                          | $(\mathbf{C} = \mathbf{b} = 1 + \mathbf{c} = 1 + 0 = 0$ |
| 0.7                              | 12                           | Deep-sea sediments; Angola Basin                                          | (Schallreuter, 1984)                                    |
| 2                                | 24                           |                                                                           |                                                         |
| 0.3 - 0.7                        | 3 - 10                       | Super-anoxic fjord; South Norway                                          | (Skei, 1988)                                            |
| -                                | 1 - 2                        | Coal basing: Pulgaria                                                     | (Kortenski and Kostova,                                 |
| -                                | 50 - 70                      | Coal basilis, Bulgaria                                                    | 1996)                                                   |
| 1                                | 10 - 15                      | Mudstone; Lower Eocene, Marquez Shale                                     | (Collins, 1982)                                         |
| -                                | 30 - 80                      | Muddy sediments (Miocene – Holocene)                                      | (Ohfuji and Akai, 2002)                                 |
| -                                | 5 - 20                       | Modern reductive sediments                                                |                                                         |
| 0.5                              | 5 - 20                       | Sulphur microbial mats; Kane Cave                                         | (Folk, 2005)                                            |
| 0.8 - 2                          | 6 - 12.5                     | Methane-derived carbonate chimneys; Gulf of
Cadiz                      | (Merinero et al., 2009)                                 |
| -                                | <200                         | Sedimentary rocks of the gold deposits (Paleozoic); Nevada, Victoria, USA | (Scott et al., 2009)                                    |
| 0.3 - 5                          | 3 - 10                       | Sediments in the South Caspian Basin                                      | (Kozina et al., 2018)                                   |

**Table 2.**

| Reagents                                                                                                                                                                           | Temperature [°C] | Duration         | Reference                    |
|------------------------------------------------------------------------------------------------------------------------------------------------------------------------------------|------------------|------------------|------------------------------|
| $FeSO_4$ , $H_2S$ , $S^0$                                                                                                                                                          | 65               | 2 weeks          | (Berner, 1969)               |
| FeSO 4 , H 2 S, CaCO 3 ; glycerine                                                                                                                | 23               | Up to 1 year     | (Farrand, 1970)              |
| FeCl 2 , H 2 S, S 0                                                                                                                               | 25, 60 or 85     | Up to 6 days     | (Sweeney and Kaplan, 1973)   |
| FeCl 2 , FeSO 4 , Fe(NO 3 ) 3 ,
Fe(NH 4 ) 2 (SO 4 ) 2 ,                                 | 25; 100          | 2 days; 4 months | (Luther, 1991)               |
| HCl, NaCl, FeS, CaSO4                                                                                                                                                              | 150 - 300        | Up to 8 weeks    | (Graham and Ohmoto,
1994) |
| Mackinawite or greigite, H 2 S,                                                                                                                                         | 70               | -                | (Wilkin and Barnes, 1996)    |
| Na 2 S, Na 2 O 3 Si, FeCl 2 ,
Fe(NH 4 ) 2 (SO 4 ) 2 , Fe(NO 3 ) 3 | 23               | Up to 2 years    | (Wang and Morse, 1996)       |
| FeS, H 2 S, KH 2 PO 4 /K 2 HPO 4 ;
Ti(III) citrate                                                                       | 60 - 100         | Up to 45 days    | (Butler and Rickard, 2000)   |

---

## Author Response (AR2)

Dear Editor,

we would like to thank you for your remarks.

We have clarified all the issues in the abstract and in the discussion. We have added additional comments and three more references.

We know that the bacteriophages studied here may not be present in sedimentation environments. However, as the study is the preliminary investigation, we chose two distinctly different bacteriophages, which are quite easy to cultivate.

We want to emphasise that Enterobacteria phage P1 belongs to *Caudovirales.* This group includes very common tailed bacteriophages that exist in nature, and thus the P1 bacteriophage was an example of that abundant group.

Furthermore, we did the metagenomic analysis of sedimentation environments in our additional project, and detected a great number of *Caudovirales* (including *Escherichia coli* phages). However, it touches on different aspects of biomineralization and will be published in another paper.

On behalf of the authors

Yours sincerely,

Paweł Działak